# Design and Synthesis of Novel Antimicrobial Agents

**DOI:** 10.3390/antibiotics12030628

**Published:** 2023-03-22

**Authors:** Zeinab Breijyeh, Rafik Karaman

**Affiliations:** 1Pharmaceutical Sciences Department, Faculty of Pharmacy, Al-Quds University, Jerusalem P.O. Box 20002, Palestine; 2Department of Sciences, University of Basilicata, Via dell’Ateneo Lucano 10, 85100 Potenza, Italy

**Keywords:** antibiotic, resistance, antimicrobial agents, nanoparticles, antimicrobial peptides, siderophores

## Abstract

The necessity for the discovery of innovative antimicrobials to treat life-threatening diseases has increased as multidrug-resistant bacteria has spread. Due to antibiotics’ availability over the counter in many nations, antibiotic resistance is linked to overuse, abuse, and misuse of these drugs. The World Health Organization (WHO) recognized 12 families of bacteria that present the greatest harm to human health, where options of antibiotic therapy are extremely limited. Therefore, this paper reviews possible new ways for the development of novel classes of antibiotics for which there is no pre-existing resistance in human bacterial pathogens. By utilizing research and technology such as nanotechnology and computational methods (such as in silico and Fragment-based drug design (FBDD)), there has been an improvement in antimicrobial actions and selectivity with target sites. Moreover, there are antibiotic alternatives, such as antimicrobial peptides, essential oils, anti-Quorum sensing agents, darobactins, vitamin B6, bacteriophages, odilorhabdins, 18β-glycyrrhetinic acid, and cannabinoids. Additionally, drug repurposing (such as with ticagrelor, mitomycin C, auranofin, pentamidine, and zidovudine) and synthesis of novel antibacterial agents (including lactones, piperidinol, sugar-based bactericides, isoxazole, carbazole, pyrimidine, and pyrazole derivatives) represent novel approaches to treating infectious diseases. Nonetheless, prodrugs (e.g., siderophores) have recently shown to be an excellent platform to design a new generation of antimicrobial agents with better efficacy against multidrug-resistant bacteria. Ultimately, to combat resistant bacteria and to stop the spread of resistant illnesses, regulations and public education regarding the use of antibiotics in hospitals and the agricultural sector should be combined with research and technological advancements.

## 1. Introduction

Prior to the turn of the 20th century, infectious diseases were the main contributor to high morbidity and mortality rates around the world [1]. The period of antibiotics, which saw the discovery and development of numerous antibacterial drugs, began with Fleming’s discovery of penicillin in 1929. Regrettably, the emergence of resistant strains was brought on by the overuse and careless use of antibiotics [2,3]. According to 2019 systemic analytic research, there were 4.95 million fatalities attributable to antimicrobial resistance (AMR). With 27.3 deaths per 100,000 people, western sub-Saharan Africa has the highest mortality rate, whereas Australasia has the lowest mortality rate with 6.5 deaths per 100,000 people [4]. Twelve families of bacteria have been identified by the World Health Organization (WHO) as the most dangerous to human health and have been divided into three priority groups: critical pathogens (*Acinetobacter*, *Pseudomonas*, *and Enterobacteriaceae*), high priority pathogens (*Enterococcus faecium*, *Staphylococcus aureus*, *Helicobacter pylori*, *Campylobacter*, *Salmonella* spp., *Nisseria gonorrhoeae*), and medium priority pathogens (*Streptococcus pneumoniae*, *Shigella* spp.) [5,6,7,8]. The presence of multidrug-resistant (MDR) ESKAPE pathogens (including *Enterococcus faecium*, *Staphylococcus aureus*, *Klebsiella pneumoniae*, *Acinetobacter baumannii*, *Pseudomonas aeruginosa*, and *Enterobacter* species) and extensively drug-resistant (XDR) bacteria has rendered even the most effective drugs ineffective. Therefore, its necessary to develop novel strategies and approaches to overcome the problem of increasing AMR [9,10,11]. This review outlines various strategies employed in the design and development of new antimicrobial agents. These include the utilization of nanotechnology and computational methods (such as in silico and fragment-based drug design (FBDD)) which have led to improved antimicrobial efficacy and enhanced selectivity towards target sites. In addition, antibiotic alternatives (antimicrobial peptides, essential oils, anti-Quorum sensing, darobactins, vitamin B6, bacteriophages, odilorhabdins, 18β-glycyrrhetinic acid, cannabinoids), drug repurposing (ticagrelor, mitomycin C, auranofin, pentamidine, and zidovudine) and synthesis of novel antibacterial agents (lactones, piperidinol, sugar-based bactericide, isoxazole, carbazole, pyrimidines, and pyrazoles derivatives) are novel approaches to treat infectious diseases. Furthermore, prodrugs (e.g., siderophores) and combinatorial treatments have recently been shown to be an excellent platform to design a new generation of antimicrobial agents with better efficacy against multidrug-resistant bacteria.

## 2. Antibiotic Classification

New antibiotic use methods should be implemented on a local and global level to combat resistance, and the development of novel treatments requires a thorough understanding of how antibiotics function. Table 1 summarizes how antibiotics produce their effects through a variety of mechanism of action. Antibiotic-mediated cell death is a complicated process that begins with a physical interaction between the medication and its particular target in bacteria, altering the bacterium’s biochemical, molecular, or ultrastructural levels. The development of DNA double-stranded DNA breaks, the halting of DNA-dependent RNA synthesis, cell envelop damage, protein mistranslation, and stress induction are only a few of the methods through which antibiotics can cause cell death [12]. Antimicrobial agents are divided into two categories on the basis of how they affect bacteria in a test tube: (1) bactericidal (kill bacteria) antibiotics such as β-lactams, glycopeptides, lipopeptides, rifamycins, aminoglycosides, and fluoroquinolones, and (2) bacteriostatic (prevent bacterial growth) antibiotics such as sulfonamides–trimethoprim and macrolides. Bacteriostatic substances can also be described as having a minimum bactericidal concentration (MBC) to minimum inhibitory concentration (MIC) ratio higher than four, and bactericidal substances when the MBC to MIC ratio is lower than or equal to four [13].

**Table 1 antibiotics-12-00628-t001:** List of antimicrobial agent and their mechanism of action.

Antibiotics Family	Mechanism of Action	Antibiotics
β-lactam	Binds to the serine active site of penicillin-binding proteins (PBPs) or the allosteric site in PBP2a to inhibit bacterial cell wall peptidoglycan transpeptidation [14,15].	Penicillins Cephalosporins Carbapenems Monocyclic β-lactams β-lactamase inhibitors (e.g., clavulanic acid) (Figure 1)
Glycopeptides	Interacts with the membrane-bound lipid II precursor of peptidogly and can prevent peptidoglycan from being incorporated into an essential structural cell wall component [16].	Vancomycin Teicoplanin Telavancin Dalbavancin Oritavancin (Figure 1)
Lipopeptide	Carries out their action by causing Gram-positive bacteria’s cell membrane integrity to be compromised, which results in cell death [17,18].	Polymyxins Daptomycin Amphomycin Friulimicin Ramoplanin Empedopeptin (Figure 2)
Rifamycins	RNA polymerase (RNAP) inhibitors are used to treat tuberculosis (TB) [19].	Rifampin Rifabutin Rifapentine (Figure 3)
Aminoglycoside	By attaching to the 30S ribosome’s A-site on the 16S ribosomal RNA, they inhibit protein synthesis [20].	Streptomycin Apramycin Tobramycin Gentamcin Amikacin Neomycin Arbekacin Plazomicin (Figure 3)
Fluoroquinolones	Target DNA gyrase, topoisomerase IV, and topoisomerase type II to prevent bacteria from synthesizing DNA [21].	Nalidixic acid Enoxacin Norfloxacin Ciprofloxacin Ofloxacin Lomefloxacin Sparfloxacin Grepafloxacin Clinafloxacin Gatifloxacin Moxifloxacin Gemifloxacin Trovafloxacin Garenoxacin (Figure 4)
Sulfonamides–Trimethoprim	Sulfonamides interfere with the activity of the dihydropteroate synthase enzyme by competing with *p*-aminobenzoic acid (PABA) in the process of dihydrofolate production.The dihydrofolate reductase enzyme is inhibited by trimethoprim because it competes directly with it [22].	Sulfamethoxazole Trimethoprim (Figure 4)
Macrolides	Target the nascent peptide exit tunnel (NPET) of the bacterial 50S ribosomal subunit, which is responsible for the release of newly synthesized protein from the ribosome, ultimately preventing protein synthesis [23,24].	Erythromycin Clarithromycin Azithromycin Fidaxomicin Telithromycin (Figure 4)
Tetracyclines	Bind to the small subunit’s decoding site and prevent bacterial protein synthesis [25,26].	Chlortetracycline Oxytetracycline Tetracycline Demeclocycline Doxycycline Minocycline Lymecycline Meclocycline Methacycline RolitetracyclineTigecycline Omadacycline Sarecycline Eravacycline (Figure 5)
Oxazolidinones	Block the translation sequence by interacting with the 50S subunit (A-site pocket) at the peptidyl transferase center (PTC) to inhibit protein synthesis [27].	Linezolid Sutezolid Eperezolid Delpazolid Tedizolid Tedizolid phosphate Radezolid TBI-223 (Figure 5)
Streptogramins	Inhibit protein synthesis during the elongation step by attaching to bacterial ribosomes [28]. The antibiotic has two unique structural groups (A and B) that cooperate to increase the affinity of group B in the nearby nascent peptide exit tunnel (NPET) when group A binds to the peptidyl transferase center (PTC) [29].	Quinupristin Pristinamycin Virginiamycin (Figure 6)
Phenicoles	Inhibit protein synthesis by binding to the 50S ribosomal subunit [30].	Chloramphenicol Thiamphenicol Florfenicol (Figure 6)
Lincosamides	Activate amino acid monomers by aminoacyl-tRNA, chain initiation, elongation, and termination of the formed polypeptides on the ribosome, which disrupts bacterial growth and death. These are only a few of the many processes that can be affected to prevent microbial protein synthesis [31].	Lincomycin Clindamycin (Figure 6)

## 3. Antimicrobial Resistance

The World Health Organization (WHO) describes antimicrobial resistance as a natural phenomenon that happens when germs cease responding to antibiotics that they were previously susceptible to before. Resistance makes treating infections more difficult or impossible [8,32]. In bacteria, there are two types of resistance: acquired and natural [33]. Natural resistance can either be produced or intrinsic (expressed in a species without connection to horizontal gene transfer) (the natural bacterial genes are only expressed to resistance levels after exposure to an antibiotic). Contrarily, acquired resistance can develop after acquiring genetic material that already exhibits it through horizontal gene transfer (HGT) (transformation, transposition, and conjugation) or by causing a mutation in the cell’s DNA during replication [8,33,34]. Antimicrobial resistance mechanisms include drug inactivation, decreased intracellular drug concentration, and altered drug targets (Figure 7) [35,36]. One of the most significant mechanisms of acquired resistance is the change or degradation of antibiotics. Bacterial enzymes have the ability to alter a number of antibiotics, including aminoglycoside, chloramphenicol, and β-lactam [37,38]. By increasing efflux or decreasing influx, one can lower drug concentration [39]. Bacteria can develop a high level of inherent resistance owing to this method. Porin mutations in resistant strains alter the permeability of bacterial membranes, which reduces medication uptake into the cell. For instance, OprD, a particular porin in strains of *P. aeruginosa*, can result in a mutation for carbapenem resistance [40]. The Proteobacterial Antimicrobial Compound Efflux (PACE) superfamily, the Resistance Nodulation Division (RND) family, the Small Multidrug Resistance (SMR) superfamily, the Multidrug and Toxic Compound Extrusion (MATE) superfamily, and the ATP (adenosine triphosphate)-Binding Cassette (ABC) superfamily are the six families that make up the transmembrane proteins that make up efflux pumps [41,42]. The most prevalent efflux pumps in both Gram-positive and Gram-negative bacteria are MFS and RND pumps [39]. The medication target changing is another method of resistance. Examples include the resistance to glycopeptide and polymyxin antibiotics caused by enzymes that chemically alter components of the cell membrane necessary for antibiotic binding. Methyltransferases are another example of target modifying enzymes since they change the rRNA elements on the ribosome and thus become resistant to antibiotics including aminoglycoside, lincosamide, macrolide, streptogramin, and oxazolidinone [43]. Another phenomenon known as “target protection” occurs when an antibiotic target’s resistance protein protects it from antibiotic-induced inhibition (target protection protein). Tetracycline ribosomal protection proteins (TRPPs) are an illustration of this mechanism [44].

## 4. Antibiotic Use and Resistance in Agriculture Sector

β-lactams, aminoglycosides, tetracyclines, macrolides, and other antibiotics with comparable modes of action to those used by humans are causing a lot of concern due to their possible side effects and risk management strategies [45]. Because medicines are available over the counter, their overuse, abuse, and misuse are linked to antibiotic resistance [46]. Antibiotic resistance is caused by the use of antibiotics in animals raised for food. Food safety and public health may suffer if antibiotic residues are found in products obtained from animals that are intended for human consumption [47,48,49]. Use of unnecessary antibiotics in animals for the purpose of promoting development, as well as waste products from veterinary care and livestock farming, human waste streams, and soil fertilization, can result in the release of antibiotics pollution into the environment. As a result, it is possible to think of the environment as a reservoir for antibiotics and bacteria that are resistant to antibiotics, and their resistance genes [45,50,51,52]. Reports that some bacterial infections in humans are brought on by animal pathogens (zoonotic pathogens) such as *Salmonella* spp., *Staphylococcus* spp., *Yersinia enterocolitica*, *Enterococcus* spp., *Listeria monocytogenes*, *Campylobacter jejuni* and *Escherichia coli* [53,54,55] have demonstrated that antibiotic resistance can be transmitted directly or indirectly from animal to human. To prevent antibiotic resistance and maintain the potency of the available antibiotics, a number of practices should be regulated for the prudent use of antibiotics in the clinical and agricultural sectors [56,57]. The demand for antibiotics to cure illnesses can be decreased by improving animal feed, waste management, and animals’ natural immunity. Moreover, using antibiotic alternatives including prebiotics, probiotic vaccines, and bacteriophages can help reduce the need for antibiotics [58,59,60].

## 5. Novel Therapeutic Agents

Here, we discuss the most recent methods and tools (Figure 8) used to resurrect antimicrobial drugs as a result of rising antibiotic resistance.

### 5.1. Nanotechnology in Combating Bacterial Resistance

The drug resistance dilemma can potentially be resolved with the use of nanotechnology. Nanomaterials can be metallic, semiconducting, polymer, or carbon-based and have been widely employed in studies and have been shown to be effective against infections on the WHO priority list. Antibacterial activity of nanomaterials arises from interactions between nanoparticles and bacteria, which includes cellular uptake and nanoparticle aggregation, which leads to membrane damage and toxicity. In an effort to reduce resistance, nanoparticles can serve as both antibiotics and delivery systems [61,62,63]. Metal-based NPs (such as silver, copper and gold) have drawn attention due to their different properties such as being optically active, having a large surface area, being chemically reactive, as well as mechanically strong. Favorable physicochemical characteristics of metallic nanomaterials led them to be widely used in biomedical applications [64,65]. In order to circumvent vancomycin resistance, a short half-life, and the necessity for a greater dose, Yadav et al. [66] used a nanosized vehicle system for delayed and sustained release of the antibiotic. Vancomycin was captured using arginine-α,β-dehydrophenylalanine nanospheres in the procedure. When compared to vancomycin alone, which only moderately inhibited *S. aureus* growth in in vitro and in vivo tests, nanoparticles effectively inhibited *S. aureus* growth. The antibacterial therapy holds out a lot of promise thanks to the delivery mechanism. A sliver nanoparticle conjugated with chitosan was created by Mohammadinejat et al. [67] and tested for antibacterial and antibiofilm properties against carbapenem-resistant *Acinetobacter baumannii* (CRAB) and methicillin-resistant *S. aureus* (MRSA). The results showed that, compared to chitosan (64, 16 μg/mL) and Ag Np alone (32, 16 μg/mL), the nanoparticle conjugate had a MIC90 of 8 μg/mL against CRAB isolates and 4 μg/mL against MRSA. Ag Np–chitosan conjugation is a successful alternative with antibacterial and anti-biofilm effects against CRAB and MRSA isolates, as demonstrated by the conjugate’s ability to reduce biofilm formation of CRAB and MRSA isolates by ¼ MIC concentration (2 μg/mL) and ½ MIC concentration (2 μg/mL), respectively. Polydopamine (4-(2-aminoethyl)benzene-1,2-diol, PDA) has strong hydrophilicity and biocompatibility, and Na Xu et al. [68] designed ferrous sulfide-polydopamine nanoparticles (PDA@FeS NPs) in which ferrous and sulfur ions preserve normal body physiology. The photothermal antibacterial activities of PDA@FeS NPs were very effective against both *E. coli* and *S. aureus*. When hydrogen peroxide is present, the near-infrared (NIR) light-mediated release of ferrous ions under weakly acidic conditions (approximately 26.5%) caused the generation of harmful hydroxyl radicals (.OH), which caused bacterial cell membrane damage and content leakage. PDA@FeS NPs’ germicidal properties offer a fresh approach for creating new antibacterial platforms. Moreover, Li et al. [69] created a positively charged porphyrin iron-based porous organic polymer called FePPOP_Hydantoin_ that generates a significant quantity of hydroxyl radicals. FePPOP_Hydantoin_ provided strong near-infrared (NIR) absorption, superior stability, high density of surface catalytic centers, and reproducibility. In addition, it gave a multi-amplified antibacterial efficacy by combining photo-Fenton and peroxidase mimetic catalytic treatment. The efficiency of the nanozymes against bacterial infection was demonstrated in an in vivo study using infected mice with *S. aureus*. Furthermore, at the nanoscale range, diamagnetic silver nanoparticles (Ag) are renowned for their antiviral, anticancer, and antibacterial effects. El-Bassuony [70] investigated the results of combining ferromagnetic (cobalt) and paramagnetic (copper) components with silver-magnetite nanoparticles. The results demonstrated that, compared to copper nanoferrite (CuAF), cobalt nanoferrite/silver-magnetite (CoAF) nanocomposites produced a more dramatic effect in the magnetic experiments. This is due to the higher coercivity Hc and saturation magnetization Ms of CoAF. Both nanocomposites shown robust antimicrobial activities when tested against Gram-positive and Gram-negative bacteria, indicating the potential application of nanomaterials as antibacterial agents. Human alpha-defensin 5 (HD5) antimicrobial peptide (AMP) was coupled with myristic acid (tetradecanoic acid) to produce Myristoylated HD5 nanobiotics in a method utilized by Lei et al. [71] to induce self-assembly. Cationicity and hydrophobicity are both critical for killing Gram-positive and Gram-negative bacteria. Human alpha-defensins clustered in nanoassemblies combine these two characteristics for efficiently killing bacteria. Furthermore, the nanobiotic showed resistance to proteolytic degradation in vivo, and minimization of renal excretion due to increased molecular size which led to the improvement of the bioavailability of the HD5-myr nanobiotic. The self-assembled nanobiotic shown improved broad-spectrum bactericidal action specifically against *E. coli* and MRSA by disrupting the cell wall and possibly other membrane structures, according to in vitro data. Excellent tolerability was also demonstrated in in vivo testing, where it was discovered that the nanobiotic could protect mice from MRSA skin infections and save them from *E. coli*-induced sepsis [72].

#### Implementation of Quality by Design (QbD) Approach in Nano-Delivery

The phrase “quality by design” (QbD) refers to products that are free of contamination and provide the consumer with the therapeutic benefit stated on the label. The FDA promotes the application of QbD concepts in the creation, production, and oversight of pharmaceutical products. The QbD technique strengthens development capability, speed, formulation design, and the manufacturer’s capacity to pinpoint the underlying reasons of manufacturing failures, which increases the effectiveness of product development and production. It also infuses quality into the product itself [73,74]. By employing a methodical QbD technique, Joshi et al. [75] created rifampicin-loaded bovine serum albumin nanoparticles (RIF-BSA-NPs) with optimal particle size and entrapment effectiveness for useful intravenous use. The 72 h sustained release of rifampicin from the BSA NPs matrix indicates that the formulated NP is appropriate for intravenous administration with the potential to enhance rifampicin’s therapeutic effects. The sulfonamide antibiotic Silver Sulphadiazine (Silver [(4-aminophenyl)sulfonyl](pyrimidin-2-yl)azanide, SSD), which is an efficient antibacterial drug used for treating burn wounds, was examined by Thakur et al. [76] using the QbD paradigm. The penetration and retention efficacy of the medicine was increased by SSD in an organo-gel foundation based on egg oil, according to the results. Furthermore, the prolonged release of the medication from the body decreased the need for frequent application and the creation of scars, which improved patient compliance. Because of this, SSD egg oil organogel holds promise as a delivery strategy for the treatment of burn wounds and associated infections. Systematic QbD was utilized by Ghodake et al. [77] to create a dry powder inhaler of sodium cefoperazone (**1**, Figure 9) based on liposomes. The effectiveness of liposomes as an anti-biofilm agent against *P. aeruginosa* was examined. The first crucial step in quality-based development is defining the target product profile (TPP), which is a summary of a drug product’s quality attributes including the route of administration, dosage form, pharmacokinetic parameters, and others. The final dosage form was a dry powder which was filled into capsules to be administered via pulmonary inhalation. Particle size and % entrapment efficiency were also selected as important quality attributes (CQAs) to guarantee the quality of the finished product. When compared to the free drug, the cefoperazone liposomal formulation demonstrated stability, nanometer-sized particle size, high drug entrapment, and enhanced in vitro antibacterial, antibiofilm, and eradication at almost 1 g/mL. For the formulation to be employed for the treatment of *P. aeruginosa*-related lung infections after cystic fibrosis, more testing is required. For the development of drugs and reliable quality control, a thorough understanding of QbD-based processes and design spaces is crucial. These spaces correspond to the critical process parameters (CPPs). A complex liposomal amphotericin B (AmB) product was created by Liu et al. [78] and improved utilizing the QbD technique. Research has demonstrated that the manufacturing technique, formulation ingredients, and curing temperature applied during the production process—and more specifically, the hydration and microfluidization—have a significant impact on the drug’s efficacy. Moreover, the design space was investigated for the reliable creation of a therapeutic product with undesirable qualities. The broad design space was observed at higher curing temperatures, lower di-stearoyl-phosphatidylglycerol (DSPG)-to-phospholipid ratios, and greater active pharmaceutical ingredient (API) to phospholipid ratios. Although AMPs offer a viable alternative to antibiotics, problems with toxicity and low bioavailability still need to be resolved. To get beyond these obstacles, Qbd-based antimicrobial peptide modification and formulation design are presented by Manteghi et al. [79] in order to create a more stable, affordable, and effective delivery to the target location. In addition to the immune system’s manufacture of antibodies, it was discovered that PEGylation of AMPs causes a decrease in biological activity due to the loss of its positive charges. The possible dangers associated with the AMP PEGylation process were examined using the AMP model PGLa (GMASKAGAIAGKIAKVALKAL-NH2). The parameters used to prioritize rankings included final size, conjugate activity, and specificity. The development of an optimum formulation of PGLa for a possible drug delivery system results from taking into account all the important criteria and choosing the best procedures and materials. As a result, the early pharmaceutical developments integrated by QbD technique aids researchers in risk-based product optimization. Understanding the cause-and-effect relationships during the initial Risk Assessment (RA) process provides the foundation for the experimental design and development support to obtain the final product within the desired quality range.

### 5.2. Computational Methods in the Development of New Antibacterial Agents

#### 5.2.1. In Silico Modelling

In silico modeling is a term used to describe computer-assisted experimentation that combines the benefits of both in vivo and in vitro research. Almost limitless parameters can be used in in silico models to provide knowledge that is not ethically or practically possible to gather *via* conventional methods. In medicine and therapies, computational models or simulations enable predictions and lead to discoveries [80,81]. Wet-lab and in silico approaches are useful in the quick identification of new lead AMPs candidates, according to Oyama et al. [82] The rumen metagenomic dataset was used to perform prediction and similarity analysis of AMPs. Among 829 sequences, six AMPs were found and demonstrated to have potential action. Two of these six possibilities, designated as HG2 (MKKLLLILFCLALALAGCKKAP) and HG4 (VLGLALIVGGALLIKKKQAKS) for additional characterization, were found. With negligible toxicity to human primary cell lines, experimental evaluation and characterization of HG2 and HG4 indicated antibacterial efficacy against Gram-positive bacteria. The suppression of other cellular processes and contact with the cytoplasmic membrane of target cells are examples of possible mechanisms of action. These peptides may be used safely as alternative treatments with antibiofilm action for the treatment of bacterial infections because of their non-toxic effect and in vivo efficacy against MRSA USA300 infection in the *Galleria mellonella* infection model. With the purpose of discovering less expensive antibacterial therapeutic medications, a study by Masalha et al. [83] combined a set of 628 antibacterial pharmaceuticals with active domains and 2892 natural products with inactive domains. A highly discriminative model was built using the iterative stochastic elimination (ISE) technique to index natural materials for their antibacterial bioactivity. Ten natural compounds scored highly as potential antibacterial medication candidates, allowing virtual screening to identify 72% of the antibacterial medicines. Caffeine and ricinine (**2** and **3**, respectively, in Figure 9) were the two molecules among ten natural products that were found and tested for their antibacterial action. To speed up the process of finding new drugs, the precisely designed prediction model can be used for virtual screening of enormous chemical databases. In the AfroDb database, there are more than 16,000 African plant structures that have well-calculated ADMET attributes. Alhadrami et al. [84] investigated the possible antibacterial action of medicinal plants from North Africa, particularly against the D-alanine-D-alanine ligase (Ddl-B) or DNA gyrase B subunits (Gyr-B) or both of the *E. coli* enzymes. The crystal structures of *E. coli* Ddl-B and Gyr-B served as the basis for the beginning of structural-based virtual screening. The top-scoring hits were anthraquinones (**4**, Figure 9), and their efficacy against Ddl-B, Gyr-B, multidrug-resistant (MDR) *E. coli*, MRSA, and VRSA was examined in vitro. Some of the tested derivatives, including emodin and chrysophanol (**5** and **6**, respectively, in Figure 9), demonstrated strong micromolar enzyme inhibition as well as antibacterial activity against the bacteria in question with MIC values ranging from 2–64 μg/mL and low to moderate cellular cytotoxicity. These discoveries represent an important step in the creation of new antibacterial drugs to combat MDR strains. In silico molecular docking was employed in Ali’s study [85] to examine ten naturally occurring marine fungus derived chemicals against a mutant enzyme from *Neisseria gonorrhoeae*. The SWISS-ADME database investigated the chemicals to determine their non-toxicity. Elipyrone A (**7**, Figure 9) with six hydrogen bonds was the best compound when the compounds’ binding affinity, chemical interactions, and toxicity were examined. Speck-Planche et al. [86] constructed a multitasking model for quantitative-structure biological effect relationships to study anti-*Pseudomonas* activities and ADMET features of organic compounds (mtk-QSBER). To evaluate the created model, delafloxacin (**8**, Figure 9) was employed as a case study. The drug’s outstanding similarity to experimental testing was revealed by the results, confirming the model’s value for virtually screening anti-*Pseudomonas* medicines. Thymoquinone (TQ), a phytoconstituent of *Nigella sativa* essential oil with possible antibacterial activity, was investigated by Qureshi et al. [87]. TQ was molecularly docked in silico against a number of antibacterial target proteins. *S. epidermidis* ATCC 12228 and *Candida albicans* ATCC 10231 were the bacterial and fungal strains that were most vulnerable to TQ. N-myristoyltransferase from *Candida albicans*, NADPH-dependent D-xylose reductase from *Candida tenuis*, D-alanyl-D-alanine synthetase (Ddl) from *Thermus thermophilus*, and transcriptional regulator qacR from *S. aureus* are the four preferred target proteins for TQ as determined by in silico molecular docking. The most effective binding, according to molecular dynamics (MD) simulations, was between TQ and Ddl or NADPH-dependent D-xylose reductase. The study emphasizes TQ’s promising effectiveness against multidrug-resistant (MDR) infections, particularly *Candida albicans* and Gram-positive bacteria. A total of 45 strains of the zoonotic and foodborne pathogen *Aliarcobacter butzleri* (*A. butzleri*) were examined by Müller et al. [88] utilizing the gradient strip diffusion method and whole-genome sequencing for antibiotic susceptibility testing. Erythromycin, doxycycline, tetracycline, ciprofloxacin, and streptomycin resistance were found in German strains. Possible resistance mechanisms were identified using in silico resistance profile prediction which utilized a specially created database (ARCO IBIZ AMR). GyrA point mutation and ciprofloxacin resistance have a strong link, and ampicillin resistance and bla3 gene have a weaker correlation. Moreover, in silico virulence profiling revealed a whole lipid A cluster presents in all examined *A. butzleri* genomes as well as a broad spectrum of putative virulence markers.

#### 5.2.2. Fragment-Based Drug Design (FBDD)

Fragment-based drug design (FBDD), whose library contains thousands of fragments, is a potent method for creating lead compounds. Novel and potent inhibitors of *Mycobacterium tuberculosis* (Mtb) enzyme 2-*trans*-enoyl-acyl carrier protein reductase (InhA) was designed using FBDD by Sabbah et al. [89] Using Differential Scanning Fluorimetry (DSF), nuclear magnetic resonance (NMR), and X-ray, 18 hits were found after screening a library of 800 pieces. NMR and X-ray techniques were used to confirm the found hits. Although the fragment hits had no discernible inhibitory activity, molecular docking and the fragment-growing technique allowed for the development of effective and new InhA nanomolar inhibitors. With submicromolar IC50 values, the insertion of a benzothiophenene resulted in the synthesis of powerful inhibitors of InhA, such as the *N*- [3-(aminomethyl)phenyl]-5-chloro-3-methylbenzothio-phene-2-sulfonamide (**10**, Figure 9). Notwithstanding the actual minimal inhibitory/bactericidal concentrations, it is important to investigate additional factors including resistance, stability, and ADME characteristics. According to the study, FBDD is a useful method for creating novel inhibitors [89,90]. Several Mtb enzymes are also being explored as potential targets for developing new drugs, such as Decaprenylphosphoryl-β-D-ribose 2′-epimerase (DprE1), which is involved in the metabolic pathway responsible for cell wall structure [91,92]; β-ketoacyl-AcpM synthase (KasA), which is crucial for fatty acid biosynthesis; a transcriptional repressor (EthR), Antigen 85 (Ag85), which is involved in the mycolic acid synthetic pathway; and 7,8- diaminopelargonic acid synthase (BioA), which play a vital role in the biosynthesis of the biotin pathway and arginine biosynthesis pathway [91]. DprE1 inhibitors have been identified as piperidinylpyrimidine derivatives (**11**, Figure 9) by Borthwick et al. [93] during initial screening utilizing SAR with MIC90 values of 30.6 μM and 15.6 μM. The newly found substance has shown positive in vivo results against acute Mtb. FBDD was utilized by several scientists to identify hit compounds for these targets. Thiolactomycin (TLM) (**12**, Figure 9) and pantetheine analog (PK940) (**13**, Figure 9) were created as KasA enzyme inhibitors by Kapilashrami et al. [94]. BDM31369, BDM31827, and 4-Iodo-*N*-prop-2-ynylbenzenesulfonamide (BDM43266); (**14**, **15**, and **16**, respectively in Figure 9), were identified by Villemagne et al. [95] as several hit compounds for EthR by library screening to treat MDR-TB. Tetrahydro-1-benzothiophene (THBTP) Analogues (**17**, Figure 9) against Ag85 were discovered by Scheich and Mendes et al. [96,97]. Through fragment screening, Dai et al. [98] discovered a powerful aryl hydrazine inhibitor of BioA 2-(aminomethyl)benzothiazole (**18**, Figure 9). The role of *ArgB*, *ArgC*, *ArgD*, and *ArgF* enzymes in the *L*-arginine production pathway in Mtb is highlighted by Gupta et al. [99], who also confirmed a hit compound against *ArgB*. The findings demonstrated that compounds NMR446 and *L*-canavanine (**19** and **20**, respectively, in Figure 9) significantly suppressed the growth of *M. tuberculosis*. The new compound *N*-(5-(azepan-1-ylsulfonyl)-2-methoxyphenyl)-2-(4-oxo-3,4-dihydrophthalazin-1-yl) acetamide (**21**, Figure 9), which was re-covered from a previous lead compound with a unique binding mode (non-conserved allosteric site), was identified by Whitehouse et al. [100] using high throughput screening (HTS) and fragment-based methods (e.g., DSF) against fumarate hydratase and Mtb H37Rv bacteria [101].

### 5.3. Antibiotic Alternatives

#### 5.3.1. Antimicrobial Peptides (AMPs)

Antimicrobial peptides (AMPs) are organic substances found in all kingdoms of life, including bacteria, fungi, plants, and animals. In addition to negatively charged microbial membranes, AMPs may also target intracellular components such as ribosomes, specific proteins, and negatively charged nucleic acids [102,103,104,105]. Protonectin (**22**, Figure 10) and polybia-CP (ILGTILGLLKSL-NH2), two antimicrobial peptides, were isolated naturally from the venom of the social wasps *Agelaia pallipes* and *Polybia paulista*, respectively. Polybia-CP and protonectin were created by Wang et al. [106,107] and demonstrated to exhibit substantial antibacterial activity against both Gram-positive and Gram-negative bacteria, including multidrug-resistant strains, by concentrating on the bacterial membrane. Campoccia et al. [108] evaluated the cytotoxicity of AMPs Dadapin-1 (GLLRASSKWGRKYYVDLAG-CAKA) on human osteoblast cells and employed it against particular bacterial species isolated from orthopedic illnesses. According to the results, Dadapin-1 significantly inhibited both Gram-positive and Gram-negative bacteria. Novel β^2,2^- and β^3,3^-bis-*homo*-ornithine/arginine peptides were created by Boullet et al. [109]. The supertryptophan residue (2,5,7-tri-tertbutyltryptophane) and cationic peptides were coupled to create AMPs with MIC values ranging from 2 to 16 µg/mL against both Gram-positive and Gram-negative bacteria. The best candidate that demonstrated a significant increase in the survival rate in vivo in septic mice was the Tbt-β^2,2^
*h* bis-Arg-OMe compound (**23**, Figure 10) [110]. Shrimp antilipopolysaccharide factors (ALFs) were employed by Matos et al. [111] to create a cysteine-free α-helix secondary structure peptide that closely follows the amino acid sequence of the central β-hairpin of *Litopenaeus vannamei* ALFs (Litvan ALF-E_33-52_ (YVNRSPYLKKFEVHYRADVK), Litvan ALF-F_31-50_ (TYFVTPKVKSFELYFKGRMT), Litvan ALF-G_35-54_ (SYSTRPYFLRWRLKFKSKVW)). The synthetic peptides’ in vitro results showed a wide range of activity against both Gram-positive and Gram-negative bacteria and fungus, as well as their capacity to operate synergistically, which highlights the possibility for discovering new drugs. Because of the toxicity concerns and pharmacokinetics restrictions associated with AMPs, Zharkova et al. [112] studied the synergistic effect of a number of AMPs (protegrin 1 (PG-1) (RGGRLCYCRRRFCVCVGR), bactenecin ChBac3.4 (RFRLPFRRPPIRIHPPPFYPPFRRFL-NH2), and RFR-ChBac3.4 (RFRRFRLPFRRPPIRIH-NH2)) with antiseptic agents (sodium hypochlorite, etidronic acid, dioxydin, poviargolum, prontosan) and some surfactants (amphoteric cocamidopropyl betaine and anionic sodium lauroyl sarcosinate) against resistant bacteria and biofilms and whether the toxic side effects can be reduced toward the host. The combination of AMP, antiseptic, and surfactant showed promise, with improved effectiveness against the formation of biofilm and lower toxicity for topical formulations. For the careful and effective development and preservation of AMPs-based products, further research needs to be conducted. AMPs from predatory myxobacteria were studied in silico by Arakal et al. [113]. Myxo_mac104 (VNRVTRVIATRRNEAERIGVPLYF), Stig_213 (VVKTVVSRAYTRAGLAQRLGWHDLRHSTRT), Coral_AMP411 (MMGAPTRRFKHHAWHETTVARRATARYVGGLSSRFVTR), and So_ce_56_913 (VEKSEKAISGARRG-SPIVNRHVVHLEHVRLKGPYRLSDRLSSAPRTSTRV) were used to create the four AMPs. The in vitro results of So_ce_56_913 and Coral_AMP411 revealed significant MIC values and revealed that, in addition to their anti-inflammatory and antifungal activities, most putative AMPs are active against more than two bacterial pathogens, with negligible activity against viruses. Myxobacteria-derived AMPs could be a viable source of new antibacterial compounds. Sp-LECin (GCVFLLPAKPHNYKKVFLSKGV), a C-type lectin homolog containing 22-amino acids from *Scylla paramamosain*, was created by Chen et al. [114]. In addition to disrupting microbial membrane integrity and causing a leakage of cellular contents, Sp-LECin was discovered to have antibacterial and anti-biofilm activity against *Pseudomonas aeruginosa*. This is because Sp-LECin binds with lipopolysaccharide to increase membrane permeability, which causes the production of reactive oxygen species (ROS), which kills *P. aeruginosa*.

#### 5.3.2. Essential Oils

Although there is a lack of clinical proof, plant essential oils (EOs) are harmless, fragrant, oil-like volatile components that are utilized as a natural therapy for the treatment of different chronic conditions. Although EOs also exhibit strong antibacterial activity, their precise mode of action is unclear, which has restricted their use [115]. Using gas chromatography-mass spectrometry (GC-MS), Tofah et al. [116] investigated the biochemical and antimicrobial activity of lavender, *Lavandula multifida* L. They also identified the essential oils extracted from *L. multifida* and discovered that the main component is camphor (**24**, Figure 10) in addition to other extracts like 1,8-cineole and alpha-pinene (**25** and **26**, respectively, Figure 10). *E. coli* and *S. aureus* were resistant to the antibacterial effects of *L. multifida* essential oil. Su et al. [117] investigated the chemical make-up, antibacterial, and antioxidant properties of the essential oil from *Centipeda minima* (EOCM) (trans-chrysanthenyl acetate, thymol, aromadendrene, and β-caryophyllene), (**27**, **28**, **29** and **30**, respectively, Figure 10) as well as the two monomers thymol and carvacrol (**31**, Figure 10). The three chemicals (EOCM, thymol, and carvacrol) were found to have strong antibacterial activity because of their effects on bacterial cell membranes, which result in material loss, as well as their suppression of protein and biofilm development, all of which hinder bacterial normal growth. *Psidium guajava* (guava) leaf essential oil (PGLEO) (limonene (**32**) and β-caryophyllene (**30**)) (Figure 10) was studied by Alam et al. [118] for its potential to treat oral infections and oral cancer. PGLEO was found to exhibit potential antibacterial action against *Streptococcus mutans* (*S. mutants*) and *Candida albicans* (*C. albicans*) in vitro and in silico studies. These findings make PGLEO a valuable source for the development of novel therapeutic agents to treat oral infections. Cinnamon essential oil (CEO), which is a secondary metabolite derived from dried cinnamon, was examined for its antibacterial properties by Zhang et al. [119] To determine their mode of action, CEO with cinnamaldehyde as the main component (**33**, Figure 10) was evaluated on *Salmonella Enteritidis* (*S. enteritidis*). According to the results, CEO decreased bacterial metabolism by inhibiting ATP, ATPase, and the tricarboxylic acid cycle (TCA), which had an impact on *S. enteritidis’s* ability to breathe.

#### 5.3.3. Anti-Quorum Sensing (QS)

One of the primary mechanisms used by bacteria to resist antibiotics is the creation of biofilms. Biofilm development is regulated by bacterial chemical communication (quorum sensing) using auto-inducers (AIs) from the *N*-acyl homoserine lactones (AHLs) group [120,121]. A potential treatment strategy to reduce bacterial virulence factors and combat antibiotic resistance is to target QS. A naturally occurring substance derived from *Hamamelis virginiana* called hamamelitannin (2′,5-di-O-galloyl-d-hamamelose, HAM) (**34**, Figure 10) interferes with *S. aureus’s* QS by altering the biofilm’s susceptibility to vancomycin via the trapP receptor [122,123]. The 5-*ortho*-chlorobenzamide derivative (*N*-(((2R,3R,4S)-4-(benzamidomethyl)-3,4-dihydroxytetrahydrofuran-2-yl)methyl)-2-chlorobenzamide) (**35**, Figure 10) demonstrated greater effectiveness in enhancing the impact of antibiotic in vivo compared to HAM [124]. A number of pyrazole and pyrazolo [1,5-a]pyrimidine derivatives were created by Ragab et al. [125] and tested for antibacterial efficacy against Gram-positive and Gram-negative bacteria that were multidrug resistant. Five substances (ethyl 4-((3,5-diamino-1H-pyrazol-4-yl)diazenyl)benzoate (**36**), ethyl 4-((5-amino-3-((4-(dimethylamino)benzylidene)amino)-1H-pyrazol-4-yl)diazenyl)benzoate (**37**), ethyl-4-((2-amino-5,7-dimethylpyrazolo [1,5-a]pyrimidin-3-yl)diazenyl)benzoate (**38**), ethyl-4-((2,7-diamino-6-cyano-5-(4-(dimethylamino)phenyl)pyrazolo [1,5-a]pyrimidin-3-yl)diazenyl)benzoate (**39**), and ethyl-4-((2-amino-6-cyano-5-(4-(dimethylamino)phenyl)-7-hydroxypyrazolo [1,5-a]pyrimidin-3-yl)diazenyl)benzoate) (**40**) (Figure 10) were discovered to exhibit significant antibacterial activity through biofilm inhibition especially against *S. aureus* and *P. aeruginosa*, making these compounds promising anti-QS candidates and lead molecules in drug discovery. A brand-new small molecule called ML364 (**41**, Figure 10) was found by Zhang et al. [126] which acts on the synthesis of staphyloxanthin and pyocyanin in *P. aeruginosa* and *S. aureus*, respectively. Results obtained in vitro and in vivo demonstrated that ML364 interfered with pathogens’ QS systems by preventing the detection of AI-2 or its nonborated form (S)-4,5-dihydroxypentane-2,3-dione (DPD) signaling, which highlights the development of novel antibacterials for the treatment of resistant bacteria. Moreover, Hurtová et al. [127] synthesized a number of halogenated compounds that target the bacterial QS protein AI-2 and examined their biological effects on *S. aureus* and *P. aeruginosa*. In particular, the brominated derivatives of flavonoids highlight the promising action of these compounds as antibacterial agents, but further toxicological and pharmacological tests should be conducted. Flavonolignans (silybins AB (**42**, **43**), silychristin A (**44**)) (Figure 10) was used to prepare the halogenated derivatives (**45**, Figure 10), which showed to exhibit an inhibitory action on the adhesion of bacteria to the service in addition to preventing biofilm formation.

#### 5.3.4. Vitamin B6

Vitamin B6 plays a crucial function as a possible antibacterial agent against *Acinetobacter baumannii*, according to Nimma et al. [128]. Enzymes of the vitamin B6 biosynthesis pathway are only present in bacterial pathogens and are not present in human hosts, making them potential therapeutic targets. The first step in the biosynthesis process for vitamin B6 is the conversion of D-erythrose-4-phosphate (E4P) to 4-phosphoerythronate, which is carried out by the enzyme erythrose-4-phosphate dehydrogenase (E4PDH). The enzyme additionally facilitates the transformation of glyceraldehyde-3-phosphate (G3P) into 1,3 bisphosphoglycerate (1,3BPG) [128,129]. The research demonstrated that E4PDH operate as cell surface receptors for the human iron transport proteins lactoferrin (Lf) and transferrin (Tf). Given its two essential functions in metabolism and iron acquisition, the E4PDH enzyme from *A. baumannii* may be critical in bacterial pathophysiology [128]. In addition to vitamin B6 derivatives, metals such as gold [130], nickel, copper [131] and gallium [132] are employed. Metal complexes were investigated for their potential antibiotic efficacy against various bacterial strains, and the results against resistant bacteria were encouraging. This provided a basis for further research into new antibiotic classes.

#### 5.3.5. Bacteriophages (Phages)

Bacteriophage viruses can be utilized specifically to fight bacteria as a result of the rise in multidrug resistance microorganisms. Several infection cycles exist for phages. In order to integrate their genome into the bacterial chromosome, temperate phages work to lysogenize their bacterial hosts [133]. Contrarily, obligately lytic phages inject their genome into the host cell, then command the host cell to divide, assemble into new virus particles, and burst from the host cell, resulting in its lysis [134]. Six lytic phages were utilized by Alexyuk et al. [135] against *E. coli* isolated from sewage. All *E. coli* strains’ growth was fully suppressed within 6 h by the phage cocktail. Using Sytox green, a membrane-impermeant nucleic acid dye that colors the DNA of lysed bacteria and produces a fluorescence signal as phage infection develops, Egido et al. [136] devised a technique to monitor phages infection in real time. *P. aeruginosa* and *K. pneumoniae*, two ESKAPE infections for which an increase in fluorescence indicates phage-mediated death, were subjected to the technique. The study demonstrated the value of this strategy in choosing phages for use against Gram-negative bacteria and for therapeutic purposes.

#### 5.3.6. Odilorhabdins (ODLs)

*Xenrhabdus* is capable of producing a diverse array of secondary metabolites through the use of non-ribosomal peptide synthetases (NRPSs) and polyketide synthases (PKSs) genes in both Gram-positive and Gram-negative Actinomycetes genera, as stated in [137]. By binding to a specific location on the ribosome (30S subunit), ODLs, which are cationic peptides made by NRPSs gene cluster enzymes of *Xenorhabdus nematophila*, disrupt the ability of bacteria to interpret and translate genetic code. As a result, it causes miscoding when producing new proteins, which causes bacterial cell death [138,139]. To tackle Gram-negative, Gram-positive, and multidrug-resistant bacteria, NOSO-502 (**46**, Figure 11) is a new class of bacterial ribosomal inhibitors ODLs that is safe and highly selective [140]. The effectiveness of NOSO-502 against *Enterobacter cloacae* complex (ECC), one of the main causes of nosocomial infections globally, was assessed by Pantel et al. [141] Except for two specific clusters (XI and XII) that are infrequently detected in clinical cases, in vitro results of NOSO-502 against ECC strains validated its robust antibacterial action.

#### 5.3.7. 18β-glycyrrhetinic Acid

The *Glycyrrhiza glabra* Linn. Plant, which includes many phytocompounds, as glycyrrhizin, 18β-glycyrrhetinic acid, isoflavones, and glabrin A and B, is increasingly receiving attention as an alternative to the usage of antibiotics [142]. The inert triterpenoid saponin glycyrrhizin (GA) is hydrolyzed in the body to produce the active metabolite 18β-glycyrrhetinic acid (GRA) (**47**, Figure 11), which has anti-inflammatory, antiviral, and antioxidant properties. The effectiveness of GRA against *Neisseria gonorrhoeae* was examined in vitro by Zhao et al. [143]. The results demonstrated that GRA has a strong antibacterial action with a dose-dependent reduction in viable *N. gonorrhoeae* and MICs ranging from 3.9 to 62.5 μg/mL. The way that GRA works is by preventing the development of new biofilms and reducing existing ones, which suggests that it could be an effective treatment for *gonorrhea*.

#### 5.3.8. Darobactins

Darobactins (DAR), a recently created small heptapeptide antibiotic, specifically kills Gram-negative bacteria by acting on the essential outer membrane protein (BamA) [144]. Darobactins are remarkable for their ability to attach exclusively to BamA without damaging the Bacteriodes in the human gut microflora. They are produced by the entomopathogenic bacterium *Photorhabdus khanii* [145]. The promising lead chemical DAR A (**48**, Figure 11) is produced by the biosynthetic gene cluster (BGC) that codes for the DarA precursor, the radical S-adenosylmethionine (SAM) (RaS) enzyme DarE, and the three transport-related proteins (Dar B, C, and D) [145,146,147]. Seyfert et al. [148] described biosynthetic engineering of new darobactins with increased antibacterial activity in a heterologous host. They showed that the newly developed darobactin has greater activity against carbapenem-resistant *A. baumannii* without hazardous side effects. It also binds to BamA more firmly. The effects of DAR, Polyphor peptide 7 (polymyxin B1 coupled to a cyclic peptide), [149] and a minor chemical (MRL-494) (**49**, Figure 11) [150] were all examined in vivo in *E. coli* by Peterson et al. [151]. DAR completely prevents signal binding to BamA as a result, but it has no effect on assembly during the post-binding stage. Moreover, Polyphor peptide 7 and MRL-494 may be able to block at least two steps of BamA function or inhibit OMP assembly directly, which opens the door to pairing the medications as antibacterial agents to increase the potency of the inhibition at different stages of OMP assembly.

#### 5.3.9. Cannabinoids

*Cannabis sativa* L. (*C. sativa*) has a high concentration of phytochemicals, which are what give it its therapeutic properties. The phytocannabinoids trans-Δ-9-tetrahydrocannabinol (THC) (**50**), cannabidiol (CBD) (**51**), cannabinol (**52**), cannabigerol (CBG) (**53**), and cannabichromene (**54**) are the most often characterized phytocannabinoids (Figure 11) [152]. Although the exact method by which cannabinoids damage bacterial membranes is still unknown, certain investigations have shown that this is how CBD works [153,154]. The antibacterial effects of CBD and CBG were studied by Luz-Veiga et al. [155]. In addition to inhibiting *Staphylococci* adherence to keratinocytes, the compounds also showed efficacy against *P. aeruginosa* and *E. coli*, with lethal MIC values ranging from 400 to 3180 µM. In light of this, the study recommends using phytocannabinoids as topical anti-microbial medicines for dermatological usage. The antibacterial and antioxidant properties of CBD and its homologue, 8,9-dihydrocannabidiol (H_2_CBD), were also examined by Wu et al. [156]. The outcome showed that the CBD analogue’s phenolic hydroxyl moiety is a crucial group to perform antioxidant and antibacterial activities. Consequently, because of their identical performance and time-kill kinetics curves, H_2_CBD can be utilized as a substitute for CBD. The interaction of CBD with broad-spectrum antibiotics such as ampicillin, kanamycin, and polymyxin B was studied by Gildea et al. [157]. By disrupting membrane integrity at extremely low dosages, CBD-antibiotic co-therapy showed an effective activity against *Salmonella typhimurium* (*S. typhimurium*), offering an intriguing alternative to treat *S. typhimurium.*

### 5.4. Drug Repurposing

Drug repurposing or repositioning is an approach term describing the use of approved drugs besides their original indications. This approach can bring failed drug back to life and highlight new targets and indications for existing drugs. The advantages of drug repurposing are increasing efficiency, minimizing investment and safety risks, reducing time to FDA approval, and reducing costs for pharmaceutical companies [158,159,160,161,162]. Here, we mention some of repurposed drug for antibacterial use (Figure 12).

#### 5.4.1. Ticagrelor

Ticagrelor (**55**, Figure 12) is a blood-thinning medication used to treat atherosclerotic cardiovascular disease. It works by blocking the platelet adenosine diphosphate P2Y12 receptor to prevent platelet aggregation [163,164]. According to a study by Sexton et al. [165], ticagrelor improves lung function in patients with pneumonia. This finding motivated P. Lancellotti et al. [166] to test ticagrelor and some of its metabolites (M5 AR-C133913, M7, and M8 AR-C124910) (**56**, **57**, and **58**, respectively, Figure 12) for their antibacterial activity in mouse models in vitro and in vivo [167]. The results demonstrated that against particular antibiotic-resistant Gram-positive bacteria with MBC range of 20–40 μg/mL, ticagrelor and one metabolite (AR-C124910) (**58**, Figure 12) have superior bactericidal activity to that of vancomycin. Moreover, Pant et al. [168] investigated Ticagrelor’s antibacterial and anti-biofilm activity in vivo to treat mice models with prosthetic joint infection (PJI) brought on by *S. aureus* as well as in vitro against *S. aureus* biofilm genes (icaA, icaD, ebps, fib, eno, and agr). The results revealed that ticagrelor had antibacterial and antibiofilm activity against *S. aureus* in vitro, either alone or in combination with certain antibiotics, and that it produced downregulation of biofilm-related genes, icaD, ebps, fib, and eno. Similar outcomes against *S. aureus* PJI were observed in vivo, where ticagrelor alone or in combination with cefazolin dramatically reduced bacterial concentrations on the implants by reducing bacterial dispersion to periprosthetic tissue. Hence, ticagrelor may function as an effective adjuvant therapy for *S. aureus* PJI, but additional research is required to fully understand how it works. Ticagrelor was also utilized to stop the growth of *C. difficile* by Phanchana et al. [169] using whole-cell growth inhibition assays. Data show that ticagrelor, which has a MIC range of 20–40 μg/mL against *C. difficile*, inhibits the growth of biofilms and the germination of spores.

#### 5.4.2. Mitomycin C (MMC)

Mitomycin C (MMC) (**59**, Figure 12) is a potent DNA crosslinker used for the treatment of bladder, gastric, and pancreatic cancers [170]. The drug was found to have antibacterial properties against bacterial pathogens including *E. coli*, *S. aureus*, and *P. aeruginosa* [171]. Cruz-Muñiz et al. [172] evaluated the effect of MMC on the growth of *A. baumannii* ATTC BAA-747. Results showed the ability of MMC to kill stationary-phase, biofilms and persister cells, in addition to the protection of *Galleria mellonella* larvae against lethal *A. baumannii* infection. Pacios et al. [173] combined MMC and the conventional antibiotic imipenem with the lytic phage vB_KpnM-VAC13 and tested their activity against imipenem-resistant and persister strains of *K. pneumoniae*. Phage-MMC showed synergistic effects on resistant and persister isolates, both in vitro and in vivo, whereas the phage-imipenem combination only killed the persisters, but not the imipenem-resistant isolate, which concludes that the lytic phage-MMC combination is effective against imipenem-resistant *K. pneumoniae* isolates harboring OXA-245 β-lactamase.

#### 5.4.3. Auranofin

A gold compound called auranofin (**60**, Figure 12) is used to treat rheumatoid arthritis and has been shown to have anti-bacterial and anti-biofilm activity in addition to other anti-disease activities against cancer, viral, and parasite infections [174,175]. The potential effectiveness of auranofin as an antibiotic adjuvant against carbapenem-resistant *A. baumannii* with the blaOXA-23 gene was investigated by Kim et al. [176]. Auranofin and doripenem had a synergistic effect on *A. baumannii*, which produces carbapenemase, according to the study. Along with inhibiting motility, auranofin has been shown to have anti-biofilm activity. It also altered the expression of genes related to carbapenemase biofilm and efflux pump. Auranofin and phenethyl isothiocyanate (PEITC) (**61**, Figure 12) were used to treat skin infections and have a synergistic antibacterial effect on *S. aureus*, according to Chen et al. [177]. Auranofin and PEITC treatment may operate as a promising therapy for *S. aureus* infection since the antibacterial impact of the medication combination increased with rising reactive oxygen species (ROS) and caused prevention of biofilm formation and destruction of bacterial cell structure. According to research by Hutton et al. [178], auranofin is a significant contender to treat *C. difficile* infections since it can limit the proliferation of *C. difficile* cells, as well as the formation of spores and the toxins A and B in both a mouse model and in vitro.

#### 5.4.4. Pentamidine

The bisbenzamidine pentamidine, 4,4′- [1,5-pentanediylbis(oxy)]dibenzenecarboximidamide (**62**, Figure 12) exhibits a high affinity for the DNA minor groove [179]. The antiprotozoal medication can disrupt the lipopolysaccharide-associated outer membrane of Gram-negative bacteria without interfering with their internal structure. The ability of diamidine to function as a strong adjuvant to increase the sensitivity of polymyxin-resistant bacteria to Gram-positive antibiotics in vitro is highlighted by Stokes et al. [180] Pentamidine and novobiocin together have shown a promising dose-sparing effect in vivo when treating mice with systemic *A. baumannii* infections [181]. Auranofin and pentamidine work together to combat germs that are multi-drug resistant (MDR), according to research by Yu et al. [182] (*E. coli*, *A. baumannii* and *K. pneumoniae*). The combination of non-antibiotic medications demonstrated a potent synergistic antibacterial action with increased bacterial absorption of auranofin and decreased resistance development in isolated MDR bacteria (*K. pneumoniae*).

#### 5.4.5. Zidovudine (AZT)

Zidovudine (AZT) (**63**, Figure 12) is a nucleoside analog used for the treatment of human immunodeficiency virus (HIV) infection. In infected cells the drug undergoes phosphorylation by cellular kinases and arrest viral DNA production by acting as an HIV reverse transcriptase inhibitor [183,184,185,186]. Zidovudine showed to have bactericidal activity against *Enterobacteriaceae* (*E. coli* and *K. pneumoniae*), with reported MICs of 0.01 to 3.7 μM and 0.1 to 11.6 μM, respectively. The mechanism of Zidovudine’s antibacterial activity may refer to its ability to act as DNA chain terminator after being phosphorylated by bacterial kinases [187,188]. A study by Ng et al. [189] suggested that Zidovudine can be repurposed as an oral antibacterial agent, in addition, to reduce the chances of resistance development it can be administered in combination with Tigecycline to treat carbapenem-resistant Enterobacterales (CRE) infections.

## 6. Synthesis of Novel Antibacterial Agents

### 6.1. Lactones

Lactones are highly bioactive substances with a broad range of biological activity. As the most prevalent secondary metabolites in plants, such as γ- and δ -lactones, they have key roles in communication, signaling, chemical defense, and controlling plant growth [190]. Using β-cyclocitral as a starting material, Mazur et al. [191] synthesized bicyclic lactones having a cyclohexane ring (**64**, Figure 1). Sodium borohydride (NaBH_4_) was used to react β-cyclocitral (**64**) to produce β-Cyclocitrol (**65**). The allylic alcohol (**65**) was modified by orthoacetate (Claisen rearrangement) to form the, γ, δ-unsaturated ester (**66**), and was then treated with an ethanolic solution of KOH to obtain the appropriate acid (**67**). Three halolactonization processes were performed on the acid (**67**), resulting in the production of δ-iodo-γ -lactone (**68**), δ-bromo-γ -lactone (**69**), and δ-chloro-γ -lactone (**70**). The compounds were compared to the dehalogenated tetramethyl substituted cyclohexane ring (**71**) and tested for their antibacterial and antifeedant activities against insect pests. The results demonstrated that, while dehalogenated lactones exhibited antibacterial action, halogen atoms were critical in demonstrating antifeedant activity. Purified enantiomerically hydroxyhalolactones (**73**, Figure 1), a multipurpose substance that can be employed as chiral building blocks, were produced by the hydroxylation of halolactones (**72**, Figure 1) obtained from β-cyclocitral [192]. Additional studies compared the antibacterial and antifeedant properties of halogenated and halogen-free γ-lactones (Figure 1). Tributyltin hydride was used to react iodolactones (**74**) and (**76**) to produce γ-ethyl-γ-lactones (**75**) and (**77**), respectively. Moreover, 1,8-Diazabicyclo [5.4.0]undec-7-ene (DBU) was used to dehydrohalogenate iodolactone (**74**), resulting in a combination of unsaturated lactones (**78**) and (**79**). The results demonstrated that only the γ-ethyl-γ-lactone exhibits antibacterial activity on particular strains, whereas reductive dehalogenation and dehydrohalogenation of δ-iodo-γ-lactones considerably boosted the antifeedant activity. Further research is required to assess the antibacterial activity of lactones modified with aromatic rings [193].

The antibacterial properties of γ-oxa-ε-lactones produced from flavanone were investigated by Gadkowski et al. [194]. The compounds were created through the cyclization of 2′-hydroxychalcones (**80**, Figure 2)) in the presence of sodium acetate to produce flavanones (**81**) from the reaction of 2′-hydroxyacetophenones (**78**) with benzaldehydes (**79**). Oxa-lactones (**82**) are produced when *m*-CPBA (*meta*-Chloroperoxybenzoic acid) is used to oxidize flavanones. The results demonstrated that the compound 4-phenyl-3,4-dihydro-*2H*-1,5-benzodioxepin-2-one and its methoxy group-containing derivatives (Figure 2) had an inhibitory effect on the growth of particular filamentous fungi, yeast, and pathogenic bacteria such as *Escherichia coli*, *Bacillus subtilis*, and *Staphylococcus aureus*. As a result, when lactone function was added to flavonoids, their activity increased in comparison to their parent precursor molecules.

Nafithromycin (WCK 4873) is another example of a lactone-containing substance (**83**, Figure 13). An oral lactone ketolide antibiotic called nafithromycin is used to treat respiratory tract infections such community-acquired bacterial pneumonia. Because it prevents RNA-dependent protein biosynthesis, the substance shows strong antimacrolide efficacy against macrolide-resistant *S. pneumoniae* [195,196].

### 6.2. Piperidinol

A piperidinol-containing molecule (PIPD1) (**88**, Figure 3) has been discovered to be a powerful lead agent against *M. tuberculosis* using high-throughput whole-cell screening of a large compound library [197]. Non-tuberculous pathogen *Mycobacterium abscessus* is resistant to the majority of antibiotics and disinfectants. De Ruyck et al. study’s [198] concentrated on the creation and synthesis of piperidinol derivatives (PIPD1) that specifically target the flippase activity of the mycolic acid transporter *MmpL3*. By first treating 4-piperidinone monohydrate hydrochloride (**84**) with α-Bromo-O-xylene (**85**) to produce a bromine derivative (**86**), which was then treated with *n*-butyl lithium solution to produce PIPD1 analogues, the PIPD1 analogues were created (**87**). The results demonstrated that MmpL3 inhibition can result in the pathogen’s immediate death. Moreover, it can have synergistic effects by making it easier for other chemicals such as β-lactams to get through the mycobacterium’s wall. The two steps of N-benzylation and the bromine-lithium exchange reaction were used to make PIPD1 (Figure 3). The two aromatic moieties A and B (Figure 3) and the steric hindrance of the substituent on ring B are essential for the inhibitory activity against *M. abscessus*, according to structural-activity relationship (SAR) investigations of PIPD1 analogues.

### 6.3. Sugar-Based Bactericides

All organisms use chemicals with a carbohydrate foundation for energy, and some microbial organisms can be inhibited by these substances. Gram-negative and Gram-positive microorganisms have been shown to be resistant to the broad-spectrum antimicrobial action of monosaccharide analogs [199,200]. Methyl β-D-galactopyranoside (β-MGP) (**89**, Figure 4) was reacted with 4-bromobenzoyl chloride to produce 6-O-(4/3-bromobenzoyl) (**90**, **91**), as well as its 2,3,4-tri-O-acyl derivatives (**92**, **93**), by Ahmmed et al. [201]. The antibacterial activity of the compounds was investigated in vitro and in silico. The results showed that the biological activities of the compounds were enhanced by the β-MGP structure with various aliphatic and aromatic substituents. This was supported by molecular docking, which revealed promising interactions and binding energies with bacterial and fungal proteins, making them potential antibacterial/antifungal candidates.

To evaluate their effectiveness against Gram-positive and Gram-negative bacteria in vitro, Dias et al. [202] synthesized a number of carbohydrate-based compounds derived from iso-quinoline-5,8-dione and naphthoquinone (**94**, **95** Figure 14), as well as their halogenated derivatives (**96**, **97** Figure 14). With MIC and MBC ranges of 4–64 µg/mL against Gram-negative bacteria, the compounds showed promising bactericidal activity. Only non-glycoconjugate naphthoquinones showed activity against particular Gram-positive bacteria. In a different study, Dias et al. [203] investigated deoxy glycosides as effective and selective bactericides that target PE and examined their efficacy against the indicated bacteria. Their study focused on *Bacillus anthracis* and *B. cereus* membranes that are rich in phosphatidyl-ethanolamine (PE). The results demonstrated that the deoxygenation pattern is a critical modulator for efficacy and selectivity and that the bactericidal activity was due to membrane disruption and highly permeable activity across the phosphatidylethanolamine membranes of *B. anthracis* and *B. cereus*.

### 6.4. Isoxazole Derivatives

The enteric bacterial infections, which are primarily brought on by *S. aureus* and *E. coli*, are one of the main causes of morbidity in poor nations. Amoxicillin, norfloxacin, and ciprofloxacin are the main antibiotics used to treat *E. coli*; however, they also have some adverse effects in addition to toxicity and drug resistance [204]. An important physicochemical factor that influences membrane permeability and medication absorption is lipophilicity. Isoxazoles and isoxazolines, which are heterocyclic compounds with nitrogen and oxygen, exhibit a variety of biological and pharmacological effects, including antitubulin, anti-inflammatory, antinociceptive, and anxiolytic activity. Sulfisoxazole and sulfamethoxazole, two β-lactam antibiotics, contain the five-membered heterocyclic ring known as isoxazole. Ahmad et al. [205] created isoxazole derivatives of fatty acids to prevent the growth of most pathogenic fungi, Gram-positive, and Gram-negative bacteria. Through 1,3-dipolar cycloaddition of nitrile oxide to lengthy chains of an alkene and an alkyne, the compounds were created (Figure 5). Long-chain alkynoic acids (**98**) and long-chain alkenoates (**100**) were employed as the starting materials for the synthesis of 3,5-disubstituted isoxazole (**99**) and 3,4,5-trisubstituted-4,5-di-hydroisoxazole (**101**). Using PASS software, biological activity of these chemicals was predicted, supporting their historical use as antibacterial agents. Furthermore, physicochemical data demonstrated that the majority of the compounds had drug-like characteristics, with MIC decreasing with increasing lipophilicity, a finding that suggested the compounds had antibacterial capabilities. In addition to their antibacterial properties, the compounds were also tested against yeast, and the results revealed antifungal efficacy against the tested clinical species of *Candida*. Docking studies with the 30s ribosomal subunit revealed agreement with in vitro measurements of microbial activity as well as a near proximity to the ciprofloxacin binding site. All chemicals share a similar fundamental structure; however, variable levels of interaction and binding patterns were discovered. These results could aid in the development of new antifungal medications that target microbial protein production.

The polycyclic aromatic compound acridone, which has anticancer and antibacterial properties, was used by Aarjane et al. [206] to create isoxazole derivatives. By combining two possible pharmacophores, acridone and isoxazole; nitrile oxides and *N*-propargyl acridone underwent a 1,3-dipolar cycloaddition process (Figure 6). Before being cyclized with polyphosphoric acid to create compounds (**103**), *o*-bromobenzoic acid was first reacted with *p*-toluidine or aniline to create 2-arylamino benzoic acids (**102**). Acridone derivatives were refluxed with propargyl bromide to produce 2-methyl-10-(prop-2-yn-1-yl)- acridone (**104**). Reacting with various nitrile oxides produced 3,5-disubstituted isoxazoles (**105**). Four harmful bacterial strains were used to test the compounds’ antibacterial efficacy (*Pseudomonas putida*, *K. pneumoniae*, *E. coli*, and *S. aureus*). These germs were resistant to the compounds’ moderate-to-good antibacterial activity. The isoxazole-acridone skeleton’s paranitrophenyl groups had the strongest antibacterial action against *E. coli* strains. To investigate the occupancy mode at the receptor pocket responsible for antibacterial activity, molecular docking of the drugs was carried out. The explanation for a compound’s antibacterial activity may be due to hydrogen bonds and hydrophobic interactions in the DNA topoisomerase *E. coli* receptor active site.

The parasitic infections known as leishmaniases are mostly spread by *Leishmania donovani*. For the creation of novel antileishmanial chemotherapeutics with enhanced efficacy and less toxicity, Tipparaju et al. [207] synthesized 2- [3-Hydroxy-2- [(3-hydroxypyridine-2-carbonyl)-amino]-phenyl]-benzoxazole-4-carboxylic acid (A–33853) antibiotic (**113**, Figure 7) and a number of its derivatives and were tested for biological activity. Compound (**113**), a benzoxazole natural product with strong antibacterial action, was discovered a few decades ago in a culture broth of *Streptomyces* sp. NRRL 12068. Early testing revealed that the A-33853 (**113**) substance is three times as active as miltefosine, a medicine used to treat leishmaniasis, in inhibiting the growth of *L. donoVani*, *T. b. rhodesiense*, *T. cruzi*, and *P. falciparum* cultures. With substantially lower cytotoxicity, certain analogs selectively inhibited *L. donoVani* at nanomolar concentrations. The oxidative cyclization of the base (**106**) or through the cyclization-dehydration reaction of amide (**107**) were the two methods used to create the benzoxazole intermediate (**108**, Figure 7). The amine (**109**) produced by the reduction in the nitro group in the intermediate (**108**) was linked with the 3-benzoyloxypicolinic acid (**111**) which was created from (**110**) to produce the intermediate (**112**). Antibiotic (**113**) was produced after BBr3 treatment in an excellent overall yield.

### 6.5. Carbazole

Other substances include carbazoles, which have generated a lot of attention for research into their potential as antibacterial agents. Carbazole is an aromatic heterocycle that contains nitrogen and occurs naturally and synthetically. Its ring is present in a number of drugs that have medical use, including carbazomycins and murrayafoline A [208,209,210]. Carbazoles may also be able to interact electrostatically or by intercalation to attach to DNA non-covalently [211]. 5-membered azoles can be introduced into the N-position of carbazole to suppress the growth of bacteria and fungi. Their effectiveness has been demonstrated to be superior to that of the reference medications Norfloxacin and Chloromycin. Making novel antibacterial agents has been a big topic for modification of the carbazole ring at the 3- and 6-positions [212]. The easiest way to create novel antimicrobial agents is to introduce 2-aminothiazole into the 3- and 6-positions of the carbazole backbone, as this other agent known as aminothiazole is found in many clinical antimicrobial medications, such as Cephalosporins, Aztreonam, Sulfathiazole, and others [210]. By reacting carbazole (**114**, Figure 8) with various *N*-alkyl bromides to produce *N*-alkylcarbazole intermediates (**115**), Addla et al. [210] created novel carbazole aminothiazoles (Figure 7). Halogenated carbazoles (**116**) were then produced by reacting chloroacetyl chloride with the intermediates, which were then fluxed with thiourea to create the aminothiazoles (**117**). The compounds’ antibacterial efficacy against Gram positive, Gram negative, and fungal organisms was assessed in vitro. Structure–activity relationships (SAR) revealed that the aminothiazole component was crucial for exerting biological activities, whereas the length of the alkyl chain affected the antibacterial efficacies. A carbazole aminothiazole molecule with a minimal inhibitory concentration (MIC) of 4 µg/mL, which outperformed the reference medications chloromycin and norfloxacin, showed negligible cytotoxicity to Hep-2 cells and good antibacterial activity against MRSA. Moreso, it was discovered that the substance interacts with DNA *via* hydrogen bonds and electrostatic interactions, which may prevent DNA replication and, as a result, have antimicrobial effects.

An innovative series of carbazole compounds was created by Xue et al. [213] (Figure 8). The chemicals were first made by *N*-alkylating or arylating carbazole (**114**), which produced intermediates (**118**). Compounds (**118**) and (**119**) were then put through a formylation reaction (Vilsmeier–Haack) to produce 9-substituted carbazole-3-carbaldehyde analogues (**120**) that were then reacted with four other substances—metformin hydrochloride (**121**), aminoguanidine hydro-chloride/or thiosemicarbazide (**123**), and isonicotinic moiety (**125**)—to create carbazole (**122**, **124,** and **126,** Figure 9). The substances had strong inhibitory effects against a variety of bacterial strains, including an isolate that was multidrug resistant. Dihydrotriazine group boosted antibacterial potency and decreased the toxicity of the carbazole compounds, according to SAR and docking tests. Moreover, in vitro enzyme activity studies demonstrated that the antibacterial effect of compounds binding to dihydrofolate reductase may be the cause of the effect

### 6.6. Pyrimidine Derivatives

Pyrimidine amine derivatives with two nitrogen atoms in the aromatic ring and a substituted amino outside have shown antibacterial and antiviral activities. Zhang et al. [214] synthesis and characterized pyrimidine amine derivatives containing bicyclic monoterpene moiety and evaluated their antimicrobial activity. Results showed that most of the compounds have antibacterial and antifungal activities against *K. pneumoniae*, *Streptococcus pneumoniae*, *P. aeruginosa*, *S. aureus*, *E. coli*, *MRSA*, *Bacillus cereus*, and *C. albicans.* Starting from aldehyde-ketone condensation reaction and in the presence of sodium methoxide or potassium tert-butoxide, a pinanyl ketene intermediates (**127**, Figure 10) were synthesized. Cyclization of (**127**) with guanidine hydrochloride produced Pinanyl pyrimidines (**128**). Further substitution with haloalkanes generated Pinanyl pyrimidine amines (**129**). As for camphor, condensation with p-methoxybenzaldehyde generated the intermediate (**130**, Figure 10), which was followed by cyclization with guanidine hydrochloride to afford the camphorylpyrimidine intermediate (**131**). Camphoryl pyrimidineamine compounds were finally obtained by substitution of (**132**) with alkyl.

Irrational use of pesticides leads to the emergence of drug resistance. The need for the development of a new broad-spectrum pesticide to control plant pests and disease infestations led Li et al. [215] to combine pyrimidine with sulfonate ester for the development of interesting bioactive compounds. A novel series of pyrimidine derivatives containing sulfonate esters were designed. Result showed that the synthesized compounds exhibited good antibacterial activity with certain insecticidal activity against *Xanthomonas axonopodis* pv. *Citri* (*Xac*), *Xanthomonas oryzae* pv. *Oryzae* (*Xoo*), *Ralstonia solanacearum* (*Rs*), and *Pseudomonas syringae pv. actinidiae* (*Psa*), which provided insight for the designing of new broad-spectrum pesticides. Synthesis of the compounds started from cyclization reaction using ethyl glycolate which produced the intermediate (6-methyl-2-thioxo-2,3-dihydropyrimidin4(1H)-one) methanol (**133**, Figure 11) that was converted into thioether derivatives (**134**, **135**, and **136**) by thioetherification with CH_3_I, C_2_H_5_I, and Benzyl chloride. The final compounds (**137**, **138,** and **139**) were obtained by esterification with RSOOCl.

### 6.7. Pyrazole Derivatives

Pyrazole is a nitrogen heterocyclic-containing molecule which plays an important role in the coordination chemistry with metal ions. Pyrazole-based ligands have proven to be effective for the construction of coordination complexes to be used in several fields, such as catalysis, antimicrobial activity, and in pharmaceutical and medical research [216,217,218]. Pyrazole-based coordination complexes with copper, cobalt, iron, cadmium, mercury, and nickel revealed remarkable electronic properties or catalytic selectivity and serve as efficient antimicrobial agents [219,220,221]. Therefore, new pyrazole derivatives were produced by Draoui et al. [222]. Pyrazole ligand *N*,*N*-bis(2(1′,5,5′-trimethyl-1H,1′H- [3,3′-bipyrazol]-1-yl)ethyl)propan-1-amine (L) (**140**, Figure 12) and four new coordination complexes called [Cu_2_LCl_4_] (**141**), [ML(CH_3_OH)(H_2_O)] (M = Ni (**142**), Co (**143**), and [FeL(NCS)_2_] (**144**) (Figure 12) were prepared, characterized, and their antibacterial properties were investigated against Gram-positive specimens (*S. aureus* and *Streptococcus* spp.), Gram-negative specimens (*E. coli* and *Klebsiella* spp.), and fungal spp. (*Fusarium oxysporum f.* sp. *albedinis* fungi). The ligand (L) (**140**, Figure 12) was synthesized by reacting *p*-toluenesulfonyl chloride with 2-(1′,5,5′-trimethyl-1*H*,1′*H*-*[3,3′-bipyrazol]*-1-yl)ethan-1-ol in the presence of sodium hydroxide to produce a tosyl-based derivative. Alkylation of tosylated product in acetonitrile gave a propylamine and in the presence of a base the desired product L was obtained. Furthermore, binuclear Cu(ii) mononuclear Ni(ii) and Co(ii) complexes were prepared by reacting ligand (L) with CuCl_2_·_2_H_2_O (1:2 molar ratio_4_)_2_·_6_H_2_O or Co(ClO_4_)_2_·_6_H_2_O salts (1:1 molar ratio), respectively. For the fourth complex, FeCl_2_ was first added then potassium thiocyanate (KNCS) (1:2 molar ratio) to prevent oxidation of the metal center and the prepare_2_ reacted with the L ligand (1:1 molar ratio) to yield Fe(ii) complex. The compounds showed moderate-to-decent antibacterial activity, especially complexes (**141**) and (**143**), with significant enhancement against *S. aureus* and *E. coli*., in addition to a distinguished anti-*Fusarium* activities for complexes (**141**) and (**144**).

Recently, the trifluoromethyl group (-CF3) has attracted the attention for its ability to work as a bioisostere in which it increases the lipophilicity, metabolic stability, and can alter receptor binding [223]. Alkhaibari et al. [224] designed potent antimicrobial agents by using 4-trifluoromethylphenyl-substituted pyrazole derivatives and evaluated their potency against planktonic bacteria, *S. aureus* and *Enterococcus faecalis*. The starting material (**147**, Figure 13) was synthesized by reacting 4-(trifluoromethyl)phenylhydrazine (**145**) and 4-acetylbenzoic acid (**146**). The hydrazine derivatives were prepared by reacting the aldehyde-containing pyrazole (**147**) with substituted hydrazines. The *N*,*N*-disubstituted compounds (**148**) showed potent activity in eradicating *S. aureus* and *Enterococcus faecalis* biofilms but failed to inhibit the growth of *A. baumannii*, whereas halogen-substituted hydrazones exhibited potent activity against *A. baumannii*. Antibacterial activity was found due to cell membrane-disrupting ability, but further studies are needed to analyze the mode of action and molecular targets of these compounds.

## 7. Prodrugs

Prodrugs are chemically inert compounds that the body metabolizes into active medications. They are utilized to circumvent a drug’s unstable pharmacokinetic qualities, toxicity, site specificity, and formulation issues [225]. They also help with solubility, absorption, and distribution issues. The prodrug method is utilized to change various molecular and cellular components, as well as physicochemical characteristics. It was used, for instance, to increase the oral bioavailability of a number of lactam antibiotics, including ivampicillin, talampicol, bacampicillin, and hetacillin. The effectiveness of other prodrugs, such as pyrazinamide, which is used to treat *Mycobacterium tuberculosis*, was intended to be enhanced by existing antibiotics such as ethionamide, isoniazid, and ethionamide. Prodrugs can therefore be used as a weapon to combat antibiotic resistance by either enhancing the pharmacokinetics of antibiotics that have activity against resistant pathogens, or by acting as a targeted drug to reduce host toxicity and remove resistance barriers [226]. The most current prodrugs made to combat resistance are included in the table below.

### 7.1. Siderophores

Using the bacteria’s nutrition absorption pathway to carry medications into the cells is one method for enhancing the effectiveness of antibiotics. A very promising tactic is to inject antibiotics into the bacteria through the iron absorption pathway. By using siderophores, which are low-molecular weight organic chelators (150 to 2000 Da) that are rich in the heteroatoms oxygen and nitrogen, microorganisms have evolved a highly effective mechanism for the uptake of iron. Bacteria produce siderophores, which are then released into the environment to chelate iron with an extremely high affinity [227]. In order to treat *Enterobacteriaceae* that are multidrug resistant (MDR) and carbapenem resistant (CR), including *K. pneumoniae*, cefiderocol (CFDC) (**149**, Figure 15) binds to penicillin-binding proteins, inhibiting the formation of peptidoglycan and ultimately killing the bacteria [228,229]. Using the Gram-negative bacteria’s outer membrane-located iron transport mechanism or passive diffusion, CFDC binds to ferric iron (Fe^3+^) and passes into the periplasmic region, where it detaches iron and enables the presentation of CFDC in large concentrations [230,231]. In their study of the susceptibility of *Klebsiella pneumoniae* to CFDC under iron-depleted and iron-enriched environments, Daoud et al. [229] discovered that CFDC had a susceptibility of 96.1%, which was higher than that of any other antibiotic. The findings also demonstrated that CFDC MICs increased in iron-enriched media where iron uptake receptors (*fecA* and/or *kfu*) are expressed, resulting in a loss of drug activity. In contrast, the presence of enterobactin receptors (*fepA*) is essential for capturing the drug and allowing its entry into the periplasm. As a result, with different iron acquisition mechanisms, bacteria may not be as dependent on the development of siderophores, which would reduce the uptake of catechol-CFDC. A siderophore-antibiotic compound (**150**, Figure 15) was created by Zheng and Nolan [232] by joining antibiotics with enterobactin, a tricatecholatesiderophore. Amoxicillin and amoxicillin, two β-lactam medicines, were joined to enterobactin via a polyethylene glycol linker. The combination increased β-lactam activity against *E. coli*, allowing enterobactin to carry antibiotics through Gram-negative membranes more effectively. An Enterobactin-ciprofloxacin compound (**151**, Figure 15) with an alkyl linker was created by Neumann et al. [233]. The cytoplasmic esteraseIroD enzyme, which is exclusively expressed in *E. coli*, activates the prodrug intracellularly. By employing a Gram-positive medication with three components—oxazolidinone antibiotic, cephalosporin, and bis-catechol-based siderophore conjugates—Liu et al. [234] created an elegant method to kill Gram-negative bacteria (**152**, Figure 15). The conjugates take advantage of the periplasmic β-lactamases that cleave the β-lactam ring of the cephalosporin to release the oxazolidinone, which then crosses the bacterial inner membrane and reaches its intracellular ribosomal target and kills the Gram-negative bacteria. The conjugates also take advantage of the bacterial ferri-siderophore uptake transporters for efficient passage through the outer membrane. To tackle *P. aeruginosa* species, Loupias et al. [235] created two piperazine-based siderophore mimics with catechol or hydroxypyridone chelating groups (**153**, **154**, Figure 15). Unfortunately, the complexes exhibited no antibacterial action, although they could be employed as antibiotic transporters against *Pseudomonas* species. The two compounds demonstrated a strong affinity for Fe(III) and were used to internalize gallium as a hazardous metal. A Siderophore-linked ruthenium catalyst (**155**, Figure 15) for the activation of an antibacterial prodrug inside cells was created and examined by Southwell et al. [236]. Due to its tiny size and hydrophilic nature, the fluoroquinolone antibiotic moxifloxacin (**156**, Figure 15) is mostly absorbed by bacteria through porins. The drug was derivatized at the N or C termini to provide allyl carbamate (N-moxi) (**157**, Figure 15) or allyl ester prodrugs (C-moxi) (**158**, Figure 15) to boost bacterial uptake through passive membrane diffusion in response to an increase in porin deficiency-associated resistance. Only C-moxi was used in bacterial experiments to explore its activation in the presence of a variety of siderophore-linked ruthenium catalysts against *E. coli* K12 (BW25113) due to N-poor moxi’s solubility. The results demonstrated that the combination of catalysts and prodrugs had an antibacterial impact, with the azotochelin- and dihydroxybenzoic acid-linked catalysts demonstrating the most promising cellular uptake and intracellular prodrug activation. An anticancer cisplatin prodrug (*cis*, *cis*, *trans*- [Pt(NH_3_)_2_Cl_2_(OOCCH_3_)(OH)] and l/d enterobactin enantiomer) was produced in a recent work by Guo and Nolan [237] (**159**, Figure 15). The enterobactin-mediated delivery of the platinum(IV) prodrug into the cytoplasm increased its accumulation more than ten times compared to cisplatin treatment, and the conjugate demonstrated antibacterial activity against specific strains of *E. coli* where it causes bacterial growth inhibition and filamentation. This paper reveals a siderophore attachment method for medication repurposing.

### 7.2. Carbapenem-Oxazolidinones

By combining numerous pharmacophores in one molecule, it is possible to find novel medications without having to look for new targets and may even be possible to overcome resistance. In this study, oxazolidinones, a synthetic antibacterial drug that inhibits protein synthesis by binding to the 23S RNA region, were conjugated to carbapenem, a powerful β-lactam antibiotic, *via* a thioether link to create a series of carbapenem-oxazolidinone hybrids (**163**, Figure 14). The intermediate (**162**) was first produced by coupling prepared oxazolidinonemethyl-thiols (**160**, Figure 10) with carbapenem diphenylphosphate (**161**) in the presence of diisopropylethylamine, and the hybrids were then produced by deprotecting intermediate (**162**) through catalytic hydrogenation [238].

### 7.3. Oral GyrB/ParE Dual Binding Inhibitor

Finding novel target sites is necessary due to the growing antibiotic resistance brought on by binding site mutations. For the development of dual-targeting antibacterial drugs, the ATP-binding subunits of DNA gyrase (GyrB) and topoisomerase IV (ParE) are ideal candidates [239,240]. *A. baumannii*, *P. aeruginosa*, and *K. pneumoniae* resistant strains are just a few examples of the Gram-positive and Gram-negative bacterial pathogens that can be effectively treated with a dual-targeting tricyclic class pyrimidoindole inhibitor (TriBE inhibitor) against GyrB and ParE enzymes [241]. The tricyclic pyrimidoindole small molecule compound JSF-2414 (8-(6-fluoro-8-(methylamino)-2-((2-methylpyrimidin-5-yl)oxy) -9H-pyrimido [4,5-b]indol-4-yl)-2-oxa-8-azaspiro [4.5]decan-3-yl)methanol and JSF-2659, the phosphate prodrug’s in vitro and in vivo properties, were described by Park et al. [242] (**164**, **165**, Figure 16). The prodrug JSF-2659 swiftly and completely converts to its active form JSF-2414 by host phosphatases to demonstrate significant efficacy against *N. gonorrhoeae* and its resistant strains. JSF-2659 prodrug exhibited high efficacy in decreasing microbial burdens and resistance.

### 7.4. Antimicrobial Peptides (AMPs) Prodrugs

A possible approach to reduce AMP toxicity issues and improve bacterial selectivity is to use AMP prodrugs. An anionic promoiety can be conjugated to temporarily diminish the cationic characteristic of AMPs, which can then be activated by particular bacterial enzymes. AMP prodrugs, P18, WMR, and Cephalothin-Bac8c, were created. WMR (Ac-EEEEAAAGwglrrllkygkrs-NH2) is an analog of myxinidin, P18 (Ac-EEEEAAAGkwklfkklpkflhlakkf-NH2) is a hybrid of cecropin and magainin, and Cephalothin-Bac8c is a β-lactam-AMP conjugates (**166**, Figure 16). Although Cephalothin-Bac8c is conjugated *via* a carbamate-1,4-triazole linker, P18 and WMR prodrugs were synthesized *via* amidation at C-termini and the elongation of the N-termini with amino acid AAG motif. The pro-peptides have a membrane-disrupting, broad-spectrum antibacterial effect [243,244]. The broad-spectrum antibiotic florfenicol (**167**, Figure 16) is utilized in the livestock and poultry breeding business but suffers from resistance and poor water solubility [245,246,247]. Cell-penetrating peptides (CPPs), such as polyarginine, permeate the membranes of microbial cells to suppress bacteria. A brand-new family of florfenicol-polyarginine conjugates was created by Li et al. [248]. The peptide-florfenicol conjugates were created by esterifying the hydroxyl group of florfenicol with succinic, glutaric, and hexanedioic anhydrides, followed by amidation with Argi-nine polypeptide. The substances significantly reduced the resistance of the *E. coli* strains (2017XJ30, 2019XJ20) to florfenicol and demonstrated strong action against *E. coli*, *S. aureus*, and MRSA.

Other prodrugs were recently reported to fight resistance are listed in Table 2 below.

**Table 2 antibiotics-12-00628-t002:** Prodrugs and their mechanism of actions.

Prodrugs	Mechanism of Action and Examples
Diazabicyclooctanones (DBOs)	The active drug is produced from DBOs, which are sulfate-containing prodrugs that are in vivo activated by esterase cleavage that intramolecularly assaults the electrophilic neopentyl methylene group [249]. DBOs function as strong inhibitors of class A and class C β-lactamases. The serine active site of the β-lactamase is targeted by an amide group on the five-membered ring of DBOs, forming a carbamoyl adduct. The effectiveness of the antibiotic can be restored by using the prodrug in conjunction with the proper oral β-lactam antibiotics [139,250,251,252]. Examples of DBOs are WCK 5153 (168), ANT3310 (169), and the following: avibactam (170), relebactam (171), nacubactam (172), zidebactam (173) (Figure 17).
β-Lactamase-Activated Ciprofloxacin Prodrug	A prodrug of cephalosporin and fluoroquinolone ((6R,7R) -7-Acetamido-3-(((1-cyclopropyl-6-fluoro-4-oxo-7- (piperazin-1-yl) (piperazin-1-yl) -1,4-dihydroquinoline-3-carbonyl)oxy)- methyl)-8-oxo-5-thia-1-azabicyclo [4.2.0]oct-2-ene-2-carboxylic Acid) created by Evans et al. [253] (174, Figure 17) to deliver ciprofloxacin only to bacteria that express β-lactamase. When cephalosporin is cleaved by β-lactamase, the prodrug’s 3′-cephem ester, which was created by attaching ciprofloxacin *via* a carboxylic acid, releases ciprofloxacin.
Azithromycin Prodrug CSY5669	Both an antibiotic and an immunomodulator, azithromycin. Azithromycin prodrug (CSY5669) (175, Figure 17) was created by Saris et al. [254] to enhance the immunomodulatory properties of azithromycin by combining it with nitric oxide and acetate as immune activators. It is possible to use CSY5669 as an adjuvant drug in the treatment of pneumonia brought on by MRSA by assisting in the eradication of bacteria and limiting inflammation-associated pathology. The prodrug showed an enhancement of intracellular killing of MARSA in monocyte-derived macrophages and peripheral blood leukocytes as well as reduced inflammatory responses in mice airways in vivo.
Tedizolid phosphate (TR701)	Prodrug of the antibiotic oxazolidinone tedizolid (TR701) (176, Figure 17), which is used to treat bacterial skin infections. Plasma phosphatasese converts it to its active parent drug tidezolide, which is highly active in vitro against Gram-positive bacteria, including MRSA [255,256,257].
Pretomanid	A prodrug of an antibiotic (177, Figure 17) that, after being converted to a desnitro derivative by *Mycobacterium tuberculosis* deazaflavin-dependent nitroreductase (Ddn) [258], acts by raising nitric oxide levels. To treat tuberculosis with drug resistance, it is used with bedaquiline and linezolid [259].
Ceftaroline fosamil	A prodrug (178, Figure 17) that is activated by plasma phosphatase to produce ceftaroline, which is used to treat community-acquired bacterial pneumonia (CABP) and acute bacterial skin infections [260,261].
Cephalosporin-3′-diazeniumdiolates (C3Ds) prodrugs	After reacting with β-lactamases and being broken down by transpeptidases, a nitric oxide (NO) donor prodrug with a β-lactam ring in its structure selectively releases NO. The diazeniumdiolate NO donor-containing PYRRO-C3D (179, Figure 17) is one of two C3Ds that are currently being developed. The second prodrug is DEA-C3D (180, Figure 17) which contains the phenacetyl side chain of cefaloram and the diazeniumdiolate NO donor. The prodrugs are a good possibility for lowering antibiotic tolerance linked to biofilms [262,263,264].
Triclosan glycoside prodrugs	The identification of the bacterial enzyme glycosidase resulted in the identification of glycoside derivatives as bacterium-targeting prodrugs (181, Figure 17). Gram-positive and Gram-negative bacteria are inhibited by triclosan glycoside derivatives (α-D-glycopyranosides and β-D-glycopyranosides), which has the potential to be utilized orally for the treatment of systemic infections [265,266,267]
5-Modified 2ʹ-Deoxyuridines prodrugs	The precise mechanism by which pyrimidine nucleoside derivatives work is unknown; however, some of the compounds inhibited the microbial enzyme flavin-dependent thymidylate synthase (ThyX), which is not present in humans, and others operated on mycobacterial cell wall destruction [268]. Negrya et al. [269] created carrier-linked prodrugs of 5-modified 2’-deoxyuridines (182, Figure 17) since the parent drugs, 5-dodecyloxymethyl 2’-deoxyuridine and 5- [4-decyl-(1,2,3-triazol-1-yl) methyl]-2’-deoxyuridine, were poorly soluble in water. To increase solubility, a triethylene and tetraethylene glycol moiety was linked to the 3′ and 5′ hydroxyl groups of the parent molecules using a carbonate group.
Tebipenem pivoxil Prodrug	Tebipenem pivoxil HBr salt (183, Figure 17) is a tebipenem ester prodrug that can be taken orally and has improved bioavailability. It is now being developed to treat difficult urinary tract infections in adults. It is approved for use in Japan to treat ear, nose, throat, and respiratory infections in children [270].
FtsZ-Targeting Benzamide Prodrugs	A prokaryote-specific protein called Fts-Z (Filamenting temperature-sensitive mutant Z) is involved in bacterial cell division. In order to combat methicillin-sensitive and resistant *Staphylococcus aureus* (MSSA and MRSA), PC190723 is a FtsZ-Targeting Benzamide the N-Mannich base prodrug TXY436 (184, Figure 17) was developed as a result of poor solubility; it has improved pharmacological characteristics but requires high effective doses. Because of this, a novel prodrug called TXA709 (185, Figure 17) was developed based on TXY436 with a CF_3_ group in place of the Cl on the pyridyl ring, giving it a longer half-life and higher oral bioavailability than TXY436 [271,272].
Carvacrol Prodrugs	A naturally occurring monoterpene called carvacrol can damage bacterial membranes and prevent Gram-positive bacteria from forming biofilms [273]. Carvacrol prodrugs (WSCP18-19) (186, Figure 17) were created by prenylating the hydroxyl group of carvacrol due to its low water solubility and chemical stability. The prodrugs exhibit good plasma stability, minimal toxicity, and a potential antibacterial action against *S. aureus* and *S. epidermidis* [274].
ADC111, ADC112 and ADC113	Fleck et al. [275] examined thousands of chimicals in order to find non-specific molecules that prevent alamarBlue, a viability dye, from being reduced. Three prodrugs— ADC111, an analog of the nitrofuran prodrug (187), ADC112, an analog of the tilbroquinol antimicrobial (188), and ADC113, a molecule with a di-ketone functionality that is not a member of any class of recognized antimicrobials (189)—are available (Figure 17). The prodrugs have demonstrated that they are effective in killing *E. coli* cells [276,277].
Contezolid acefosamil (CZA) prodrug	A brand-new oral oxazolidinone antibacterial medication called Contezolid (CZD) is effective against the majority of aerobic Gram-positive bacteria, including MRSA and vancomycin-resistant *Enterococcus*. The medication prevents the synthesis of 70S initiation complex, which is essential for bacterial reproduction [278]. As a result of its low solubility, the drug’s intravenous (IV) administration is restricted. Giving patients with diabetic foot infections more therapeutic options in hospitals and outpatient settings is therapy with IV administration followed by oral CZD [279]. Liu et al. [280] created the contezolid acefosamil (CZA) prodrug (190, Figure 17), an isoxazol-3-yl phosphoramidate derivative with excellent water solubility and good stability in pH conditions suited for IV delivery.

## 8. Awareness and Knowledge of Antibiotic Prescribing

Antibiotic resistance is seen as a severe concern and a global public health issue since it has increased morbidity, death, and healthcare expenditures. Antibiotic resistance was brought about by the irrational use of antibiotics in agriculture, the livestock industry, and healthcare. In addition to prescribing an inappropriate antibiotic, using antibiotics without a necessity, skipping doses, self-medicating, and sharing medications are key contributors to antibiotic resistance. This is owing to pharmacists’ lack of understanding, fear of losing clients, and lax legal protections [281,282]. Many institutions have found that implementing Antimicrobial Stewardship Plans (ASPs) can enhance therapeutic results, lower treatment-related costs, and hence slow or stop the evolution of antibiotic resistance [283]. Health professionals can also avoid hospital-acquired infections and the nosocomial transmission of MDR bacteria by following the recommendations for proper hospital disinfection and personal hygiene [284,285]. Nonetheless, both wealthy and developing nations suffer from the widespread use of antibiotics in the aquaculture and agricultural sectors to promote growth. Humans will consume antibiotics used on livestock, and resistant bacteria may spread from animals to humans, perhaps having a negative impact on human health. Moreover, antibiotics given to animals are expelled in urine and feces, which are then used as fertilizer and have an impact on the microbiome of the environment [286,287]. As a result, we must promote awareness about antibiotic resistance, enact legislation, and create global policies. In order to promote patient care and national security, the fight against antibiotic resistance requires strategies, persistent efforts, and the participation of national and international governments, healthcare professionals, industry, and the general public [288].

## 9. Conclusions

A total of 4.95 million fatalities globally are a result of the significant global public health problem known as antibiotic resistance. Through naturally occurring or acquired resistance that is created by horizontal gene transfer or DNA mutation, bacteria can rapidly reduce their susceptibility to antibiotics. In order to combat bacterial infections, new antibacterial agents must be discovered. The development of antimicrobial therapies can be facilitated by new methods for rational design and screening-based approaches, such as, nanotechnology, computational techniques (in silico and FBDD), antibiotic alternatives (antimicrobial peptides, essential oils, anti-Quorum sensing, darobactins, vitamin B6, bacteriophages, odilorhabdins, 18β-glycyrrhetinic acid, and cannabinoids), drug repurposing (ticagrelor, mi-tomycin C, auranofin, pentamidine and zidovudine), and synthesis of novel antibacterial agents (lactones, piperidinol, sugar-based bactericide, isoxazole, carbazole, pyrimidines, and pyrazoles derivatives) and prodrugs. In order to address the crisis caused by antibiotic resistance, it is necessary to coordinate efforts to revitalize research and implement new policies.

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
