# Peer review of "Design and Synthesis of Novel Antimicrobial Agents"

_antibiotics, 2023, doi:10.3390/antibiotics12030628_

Round 1

Reviewer 1 Report

The current manuscript is a thorough and reasonably complete review on strategies to circumvent the current antibiotic resistance issue. It is extensive and interesting to read, nevertheless some changes should be made before acceptance for publication:

- Strategies for combating antibiotic resistance should be summarized in a figure, namely synthesis of new molecules, nanosystems, etc.

- The use of in silico models as an advantage for new drug molecule development should be further emphasized, and quality-by-design approach to drug molecule manufacture should be mentioned, as an emerging strategy for improved efficiency;

- Drug repurposing is another efficient strategy that could be mentioned;

In what concerns the use of nanosystems:

- A comment should be added on the potential toxicity of different nanosystem types, since metallic nanoparticles are inherently more toxic that carbon based ones;

- Also the authors should specify more about the reported nanosystems, in what concerns the general advantages of encapsulating antibiotics, and also on their composition, and how it affects drug bioavailability.

Author Response

Reviewer 1

  • Comment: “Strategies for combating antibiotic resistance should be summarized in a figure, namely synthesis of new molecules, nanosystems, etc.”

 Response: A figure was added line 171 (figure8) page 18.

  • Comment: The use of in silico models as an advantage for new drug molecule development should be further emphasized, and quality-by-design approach to drug molecule manufacture should be mentioned, as an emerging strategy for improved efficiency

Response: new sections was added considering in silico models (line 299, page21) and quality by design (line 239, page 20)

  • Comment: “Drug repurposing is another efficient strategy that could be mentioned.”

 Response: new section on drug repurposing was added (line 661, page 31)

  • Comment: A comment should be added on the potential toxicity of different nanosystem types, since metallic nanoparticles are inherently more toxic that carbon based ones.

Response: A comment was added (line 177- 188 page 19)

  • Comment: “Also the authors should specify more about the reported nanosystems, in what concerns the general advantages of encapsulating antibiotics, and also on their composition, and how it affects drug bioavailability.”

 Response: Some information was added from the free available data from articles (line 188-238, pages 19, 20).

Reviewer 2 Report

I found this review article interesting for the readers and followed the journal Antibiotics’ scope based on the topic. However, the author failed to make this article more interesting to the readers as in the introduction the author didn’t summarize the topics going to be included in this review and lacked in-depth discussion. As the author didn’t discuss synthesis of all antibiotics, therefore the author should comment on this as well. The title of this review article is design and synthesis of Novel Antimicrobial agents, I was expecting more discussion on the selection of molecules and synthesis thereof.

I would recommend the article be published in Antibiotics after major corrections. 

The author needs to address the following comments/corrections.

  1.     For “Table 1. list of antimicrobial agents and their mechanism of action.”: The author should include the structure of all molecules mentioned in table 1. Before start of table 1, mentions the usefulness of table 1.

2.     For “2. Antimicrobial resistance”: Provide figure for the mode of resistance.

3.     For “3. Antibiotic use and resistance in agriculture sector”: Figure 1 needs footnotes.

4.     For: “4. Novel therapeutic agents”: Concise this part with structure in the table.

5.     For: “5. Synthesis of novel antibacterial agents: Why only few syntheses were discussed, what is rationale for choosing few molecules. For the synthesis all structures in the scheme should be similar font, yield and reaction time should be included in the reaction schemes.

6.     For:” 5. Prodrugs”: Change number 5 to 6,

7.     For: 6. Conclusions: Change number 6 to 7.

8.     Include the following references if possible.

(a)   https://doi.org/10.1016/j.bmcl.2008.05.005

(b)  https://pubs.acs.org/doi/full/10.1021/jm801241n

Author Response

Reviewer 2

  • Comment: Table 1. list of antimicrobial agents and their mechanism of action.”: The author should include the structure of all molecules mentioned in table 1. Before start of table 1, mentions the usefulness of table 1.

Response: all the structures were included from line 86, page 10, and the usefulness of table one was added (line 70, page 6)

  • Comment: Antimicrobial resistance”: Provide figure for the mode of resistance

Response:  a figure for the mode of resistance was replaced to this section (line 142, page 16).

  • Comment: Antibiotic use and resistance in agriculture sector”: Figure 1 needs footnotes.

Response:   figure1 in this section was deleted due to its wrong place and was added in line 142 page 16.

  • Comment: Novel therapeutic agents”: Concise this part with structure in the table.

Response:  a figure was added for the novel therapeutic agents and each section has its own structure line 171 (figure8) page 18.

  • Comment: Synthesis of novel antibacterial agents: Why only few syntheses were discussed, what is rationale for choosing few molecules. For the synthesis all structures in the scheme should be similar font, yield and reaction time should be included in the reaction schemes.

Response:  New synthesis materials were added (line 988 page 44 and line 1031, page 46) and yield with reaction time were added to all schemes.

  • Comment: Prodrugs”: Change number 5 to 6, 6. Conclusions: Change number 6 to 7.

Response:   Numbering was corrected (line 1081, 1252)

  • Comment: Include the following references if possible. (a)https://doi.org/10.1016/j.bmcl.2008.05.005 (b)  https://pubs.acs.org/doi/full/10.1021/jm801241n

Response:  One reference (b) was added to the article line 917, page 41.

Reviewer 3 Report

The manuscript entitled " Design and synthesis of novel antimicrobial agents" in which the authors discussed the antibiotic resistance and the development of antimicrobial therapies that can be aided by novel different classes of molecules to combat resistant bacteria and stop the spread of resistant illnesses. The work is understandable and the topic is important and interesting. The scientific narrative is well structured and flows naturally from one idea to the next.

However, this paper suffers from few shortcomings that if modified would make the manuscript very suitable for publication in Journal of Antibiotics.

Shortcomings:

1-      Please add the aim of this work in the introduction section and abstract section.

2-      Please add section of abbreviation for easy reading.

3-      Please add the funding part if the cost are paid by authors or are funded.

4-      Please paste these two sentences” Moham-madinejat et al.[60]  in line 128

and tested for antibacterial and antibiofilm properties against carbapenem-resistant in line 129”.

5-      Please delete (;) in Figure 3 legend “ Figure 3. Chemical structure of; protonectin (13)”.

Author Response

Reviewer 3

  • Comment: -Please add the aim of this work in the introduction section and abstract section Response: the aim of work was added in the abstract line 15 and in the introduction line 52
  • Comment: Please add section of abbreviation for easy reading

Response: Abbreviation section was added line 32

  • Comment: Please add the funding part if the cost are paid by authors or are funded.

Response: The funding part was deleted

  • Comment: Please paste these two sentences” Moham-madinejat et al.[60]  in line 128

and tested for antibacterial and antibiofilm properties against carbapenem-resistant in line 129”.

Response: Corrected line 195.

  • Comment: Please delete (;) in Figure 3 legend “ Figure 3. Chemical structure of; protonectin (13)”.

Response: Corrected line 509

Round 2

Reviewer 2 Report

I have re-evaluated the review article (Antibiotics-2285489) titled “Design and Synthesis of Novel Antimicrobial Agents” by Karaman and coworkers, and this review article summarized available antibiotic to combat resistant bacteria along with novel antibiotics, their route of synthesis and mechanism of actions and the author also discussed public education regarding the use of antibiotics in hospitals and the agricultural. I found this review article interesting for the readers and followed the journal Antibiotics’ scope based on the topic.

I would like to thank authors of this article for making necessary corrections as suggested by the reviewers, and the standard of this article enhanced.  

I would recommend the article be published in Antibiotics as such. 

The author needs to address the following comments/corrections.

             1.     I would prefer the abbreviation before references or in the SI.

2.     Remove “yield” from the scheme 14 as in other schemes only % was used to present the yield not the word yield for the reaction conditions. Please check the similar errors.

3.     In the reaction conditions: follow the similar pattern in each scheme (time, yield etc).

4.     Check the font size of each scheme, it should be same.

Author Response

Antibiotics-2285489-reviewers' comments’ response

Reviewer 2

  • Comment: I would prefer the abbreviation before references or in the SI.

Response: Abbreviation section was added before references line 1270, page 55.

  • Comment: Remove “yield” from the scheme 14 as in other schemes only % was used to present the yield not the word yield for the reaction conditions. Please check the similar errors.

Response:  Yield in scheme 14 was deleted line 1172, page47, and other schemes were checked.

  • Comment: In the reaction conditions: follow the similar pattern in each scheme (time, yield etc).

Response:  Similar pattern in all schemes were followed (lines 783, 829, 845, 913, 932, 1005, 1025, 1058, 1076 and 1172).

  • Comment: Check the font size of each scheme, it should be same.

Response:  Font size was checked in all schemes and it’s the same.